# Fast Human Motion reconstruction from sparse inertial measurement units considering the human shape

Xuan Xiao [1,2], Jianjian Wang[1,2], Pingfa Feng [1,2], Ao Gong[1,2], Xiangyu Zhang[1,2] & Jianfu Zhang [1,2] ✉

Inertial Measurement Unit-based methods have great potential in capturing motion in large-scale and complex environments with many people. Sparse Inertial Measurement Unit-based methods have more research value due to their simplicity and flexibility. However, improving the computational efficiency and reducing latency in such methods are challenging. In this paper, we propose Fast Inertial Poser, which is a full body motion estimation deep neural network based on 6 inertial measurement units considering body parameters. We design a network architecture based on recurrent neural networks according to the kinematics tree. This method introduces human body shape information by the causality of observations and eliminates the dependence on future frames. During the estimation of joint positions, the upper body and lower body are estimated using separate network modules independently. Then the joint rotation is obtained through a well-designed single-frame kinematics inverse solver. Experiments show that the method can greatly improve the inference speed and reduce the latency while ensuring the reconstruction accuracy compared with previous methods. Fast Inertial Poser runs at 65 fps with 15 ms latency on an embedded computer, demonstrating the efficiency of the model.

Motion capture is performed to digitally reconstruct the posture and movement of the human body. This technique has extensive applications and is commonly employed in various fields such as virtual reality (VR) and augmented reality (AR), film and game production, human factors analysis, biomechanical analysis, medical rehabilitation, and sports training.

One approach for motion capture is based on vision systems. Motion capture based on optical markers has been performed commercially due to its high accuracy[1]. Camera vision-based methods[2], such as monocular image-based[3,4], multi-view image-based[5,6], and video-based[7–9] methods are widely used. However, these methods are inevitably disturbed by occlusion and site limitations.

Although marker-based optical motion capture methods can achieve very high accuracy, they require the complex post-processing of data and may not be effective in cases with multiple occlusions. Camera-based methods offer a convenient way to estimate human motion, but they cannot guarantee effective motion reconstruction in extreme occlusion environments, and they often make erroneous estimations due to occlusions between limbs. To address occlusion[10–12] and ambiguity[13,14], some researchers have introduced a combination of inertial sensors and optical approaches[15]. While this can improve the accuracy of reconstruction to some extent, it introduces complexity in the wearing process and still imposes limitations on the range of human movements[16].

[1]State Key Laboratory of Tribology in Advanced Equipment, Department of Mechanical Engineering, Tsinghua University, 100084 Beijing, China. [2]Beijing Key Lab of Precision/Ultra-precision Manufacturing Equipments and Control, Department of Mechanical Engineering, Tsinghua University, 100084 Beijing, China. ✉e-mail: zhjf@tsinghua.edu.cn

In addition, purely inertial-based methods can be further applied in many application scenarios[17,18].

To directly use the raw outputs of Inertial Measurement Units (IMUs) to estimate the rotation of each joint of the whole body (except the fingers), dense inertial configurations are necessary. More than 18 sensors can directly estimate the rotation of each joint that affects the whole body posture without requiring excessive data post-processing. This is because the sensors are directly placed at each major joint, enabling them to directly output the orientation information of those joints[19,20]. Therefore, this method relies more on the accuracy of the sensors themselves than sparse inertial configurations, as the latter can be used to estimate joint rotations based on a large amount of prior human motion data and is not directly constrained by sensor signals. Much work is required to perform efficient filtering or nonlinear optimization to enable a sensor to obtain a high accuracy output[21–23]. Due to sufficient information from sensors, this method can provide higher accuracy and less computation, and there are already commercial solutions based on this method[19,20,24]. To eliminate the error obtained by the sensors, similar to Kalman filtering, some methods predict the next pose and calibrate the current pose using the constraints of the human motion state[24]. Zihajehzadeh et al.[17] introduced data from other Ultra-Wideband (UWB) sensors to obtain more accurate results. This kind of method involves numerous coordinate system transformations among multiple sensors, and due to the large number of sensors, it becomes challenging to ensure consistent sensor orientations before each use. As a result, a substantial amount of time is required for calibrating the coordinate system transformation between sensors and the human body before usage. In the case of sparse inertial configurations, the small number of sensors makes it relatively easier to ensure consistent sensor orientations and positions during the wearing process. The calibration process of sparse sensors is simpler than that of dense sensors.

Although widely used, this kind of method still suffers from the fact that it is highly intrusive and inconvenient considering that a large number of sensors limits the behavior of the user to a certain extent. To address this limitation, the number of sensors used can be reduced.

The sparse sensors solution faces the following challenges. First, a reduction in the number of sensors leads to the lack of joint constraints, which makes the problem underdetermined. Second, directly integrating the accelerometer measurement is unreliable. Therefore, it is impossible to directly reconstruct the human pose by a kinematic inverse solution relying on these data.

In earlier studies, researchers tended to use optimization-based methods rather than learning-based approaches. Slyper et al.[25] used 5 sensors to estimate the posture of the upper body. Since the sensors were stitched to a cloth, a strong prior assumption for pose estimation was needed. Some methods use 4–6 sensors to estimate the pose of the human body by matching the current sensor data with the recorded data[26,27]. These methods often optimize the speed of search matching by designing efficient data structures. Andrews et al.[28] constrained physical quantities such as the torque, constructed state transition equations, and reconstructed the posture of the whole body by solving convex optimization problems. Marcard et al.[29] used the motion prior to construct a nonconvex optimization problem on the entire time series and proved that the motion of the human body can be reproduced by using 6 sensors. However, these methods can only be used in scenarios with low real-time requirements.

Deep Neural Network (DNN)-based methods can effectively solve the problems of data migration and the lack of real-time performance. Huang et al.[30] proposed Deep Inertial Poser (DIP), applying deep learning to this kind of method. They used a simple bidirectional RNN[31,32] and a fine-tuning operation on a true dataset. They proposed various data processing methods, and provided an open-source dataset. Geissinger et al.[33] applied a transformer[34] with more sensors

in the layout scheme. Yi et al. proposed Transpose[35], which is used to determine whether a person is walking by introducing the probability modeling of feet touching the ground and then multiplying the leg length by the end speed to obtain the global travel speed. Jiang et al. presented Transformer Inertial Poser (TIP)[36], which used a transformer and sampling-based optimization to reconstruct motion. Yi et al. proposed Physical Inertial Poser (PIP)[37] and introduced the dual PD controller to model the torque[38–40] and optimize the output of the Transpose. However, it increases the computational burden because of the design of the optimization approach. Based on these methods, Yi et al.[41] incorporated cameras on the helmet, achieving more precise localization.

The above methods have good performance on PCs with high computing abilities. However, due to the limited signal reception distance, they are not suitable choices for real-time motion reconstruction in large-scale scenes such as streets. This is because carrying a large-volume computer can interfere with a user's original motion. Therefore, mobile terminals with limited computing power are usually preferred. The computing modules in mobile terminals, such as AR headsets, often exist in the form of embedded computers, which are compact, low-power, and have limited computational capabilities. Moreover, the methods didn't consider the body shape information that can be used. This inevitably introduces biases in body shape that require more computational effort to eliminate. Since wearing sensors requires the subjective participation of people, it is convenient to make a simple measurement of human parameters before wearing it. Our method introduces these measurements to reduce unnecessary computations.

To this end, as shown in Fig. 1, we propose Fast inertial Poser (FIP), which is a real-time motion capture method. FIP can reduce the computational burden and latency while ensuring the reconstruction accuracy, making it suitable for embedded platforms with limited computing capabilities. The pose estimation stage of Fig. 1 is divided into two stages: 1) the joint position estimation and 2) the kinematic inverse solution. This method focuses on obtaining faster performance by introducing the physical parameters of the human body shape, while the previous method did not consider such parameters. Compared to the previous methods, the main reasons for the efficiency improvement of this method are as follows: 1. eliminating additional optimization designs; 2. enhancing the expressive power of neural network structures by considering the human body shape parameters, using a kinematic inverse solver and a shared model for different sensors; and 3. removing the bidirectional propagation mechanism of RNNs.

In the position estimation stage, we use three independent recurrent neural networks (RNNs) to estimate the positions of leaf nodes and body nodes. To make the model inference process closer to the real physical process, we used the sensor-shared integral RNN to estimate the position of the leaf nodes. In other words, different sensors share the same integral RNN to estimate their displacements. Moreover independent RNNs are used to estimate the upper and lower body nodes, respectively. In addition, for each RNN, we encoded the human parameter information and inputted embedded vectors into the network.

In the kinematic inverse solution stage, since the traditional IK process is non-differentiable, we design a differentiable inverse kinematic solver based on the Skinned Multi-Person Linear model (SMPL)[42] kinematic tree. The joint positions of each frame are input to the inverse solver separately, and the output is the joint rotation.

In summary, our main contributions are as follows:

- A real-time (more than 60 FPS) motion capture DNN approach that can be run on an embedded computer is proposed.
- A design of the regression network architecture for the position of key joints with human shape inputs is introduced.

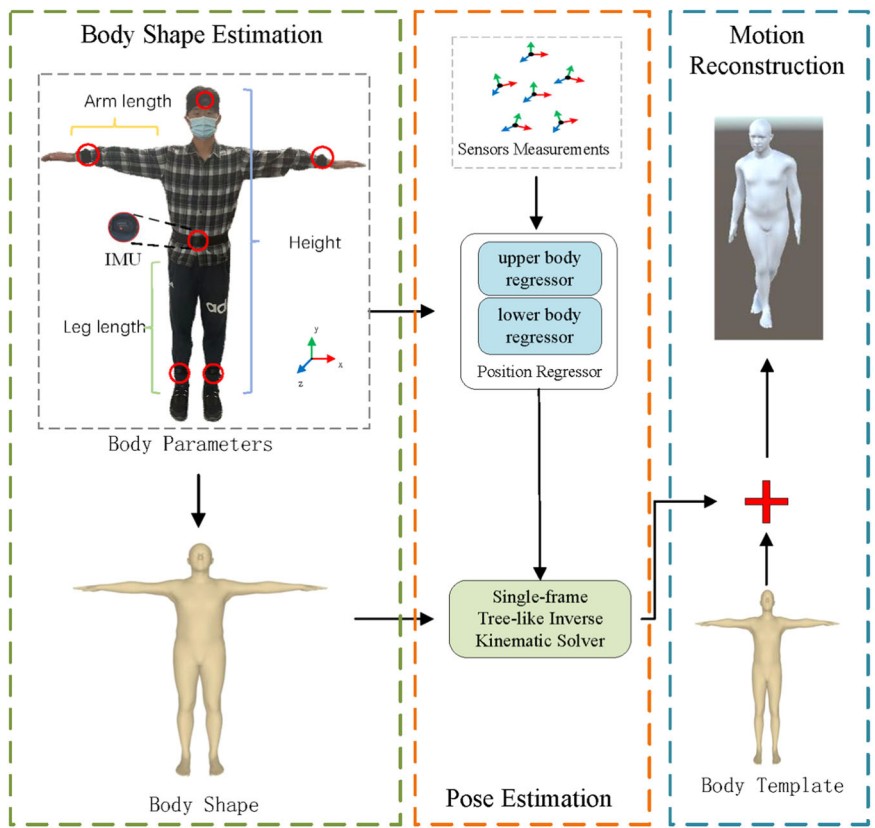

**Fig. 1 | Illustration of FIP.** FIP introduces several body parameters that are easy to measure when performing pose estimations, and then reconstructs the whole body pose from 6 IMUs. Importantly, FIP can run at 65 fps with a latency of 15 ms on an embedded computer.

- A special inverse kinematic solver based on the human kinematic tree is designed to help the model solve the joints' rotation in the current frame.

## Results

In this section, we present our experimental results. First, we compare our results with those of previous methods. Second, we perform ablation comparisons to determine whether the addition of our various submodules is necessary. Finally, we show the demo of our model.

### Datasets and metrics

We mainly use the AMASS dataset[43] and the DIP-IMU[30] dataset. The AMASS only provides human motion information, while DIP-IMU additionally provides true sensor measurement values. Since the TotalCapture[44] dataset does not provide human-related information during recording, it cannot be effectively used to verify the performance of our model. Therefore, we did not consider this dataset. The AMASS dataset contains 500 subjects, over 3000 min, and over 10,000 actions. The DIP-IMU dataset includes 10 subjects and more than 50 sets of actions. In the AMASS dataset, ground truth is obtained through optical markers, while in the DIP dataset, ground truth is iteratively optimized based on the output from dense sensors. The dataset provides real-axis-angle parameters for SMPL. With the SMPL skeleton and blend weights, it is possible to calculate the positions of the SMPL joints and mesh vertices. We collect the $\beta$ values (the shape parameters of the SMPL human body model) of various human bodies in the SMPL model from the AMASS dataset. Then, we estimate the value of the body shape parameter $\beta$ for the DIP-IMU human recorded according to the sex and height information and trained the model based on this information.

We use the following metrics to evaluate our method. The reconstruction metrics include the following: (1) *SIP Error* (SIP)

measures the mean orientation error of the upper arm and legs in the body coordinate system in degrees. (2) *Angle Error* (Ang) measures the mean angle error of the 15 key joints in the body coordinate system in degrees. (3) *Position error* (Pos) measures the mean Euclidean distance error of the joint positions in the body coordinate system in cm. (4) *Jitter* (the second derivative of velocity, $km/s^3$)[45] represents the motion smoothness performance of the model; the lower it is, the smoother and more authentic it is. (5) *Aang* is the same angle evaluation method as Transpose[35], which takes the 22 joints of the whole body into account and considers the rotation of noncritical joints to be consistent with the ground truth, and the real rotations of the SMPL joints directly provided by the dataset. (6) *Mesh* represents the mean value of the vertex error in cm. The deployment metrics include the following: (7) The time-cost per frame (TPF) measures the mean running time of each frame in ms and can be used to evaluate the model running efficiency. (8) *MEM* is the GPU memory occupation of the model. (9) *Usg* is the usage of the GPU. (10) *Latency* is the time gap between the model output and the input of the same frame. (11) *FPS (frames per second)* count the frames per second.

The TPF is tested on a graphic station, while other deployment metrics are tested on a TX2 NX card. For these metrics except FPS, lower values are better. It is worth noting that Transpose uses data of the future IMU readings as input to the model, while TIP uses data from the future frames for filtering. Therefore, their latency is longer than the inference time.

On the graphics workstations, all methods were tested using PyTorch 1.8 and CUDA 10.2. On the embedded computer TX2NX, the tests were conducted using Jetpack, PyTorch 1.8, and CUDA 10.2.

### Comparisons

Since our method uses unidirectional RNNs, there is no difference between the offline results and the online results ("offline" refers to

**Table 1 | Comparison of the performance metrics**

| Method | Reconstruction Metrics | | | | | | Deployment Metrics | | |
|---|---|---|---|---|---|---|---|---|---|
| | SIP (deg) | Ang (deg) | Aang (deg) | Pos (cm) | Mesh (cm) | Jitter ($km/s^3$) | TPF (ms) | Latency (ms) | FPS |
| DIP | 17.85 | 15.47 | 16.05 | 6.65 | 9.46 | 2.77 | – | – | – |
| Transpose | 16.69 | 11.30 | 8.86 | 5.80 | 7.34 | 0.61 | 10.6 | 120 | 27 |
| PIP | 15.02 | 10.54 | 8.73 | **4.80** | **5.95** | **0.27** | 13.3 | 76 | 13 |
| TIP | 15.40 | 10.78 | 8.95 | 5.03 | 6.33 | 0.97 | 6.5 | 127 | 23 |
| FIP(Ours) | **14.37** | **10.06** | **7.72** | 5.09 | 6.24 | 1.74 | **2.7** | **15** | **65** |

Bold denotes the best performance value for each metric.

non-real-time post-processing of data, while "online" refers to real-time processing). We compare our method with previous 6-sensor-based real-time tracking methods including DIP[30], Transpose[35], PIP[37] and TIP[36] based on the online situation and their weights are provided by open source projects in their GitHub. Due to the use of an old version of TensorFlow in the open-sourced code of DIP, which is inconsistent with other methods, we did not deploy it on the embedded device. Since PIP only uses the CPU during inference, we did not compare its deployment metrics on a GPU of the embedded computer.

As shown in Table 1, our method leads by a large margin in terms of the angle-related metrics. After simple calculations on the data in Table 1, it can be seen that even when facing the optimization-based method PIP, our approach still outperforms by 5% on SIP and Ang, and leads by 13% on Aang. Our method also maintains a 5% margin with the best approach PIP in terms of the Mesh and position accuracy. This is likely due to the optimization of the joint accelerations performed by PIP. However, our method is not good enough with regards to Jitter. On one hand, this may be due to the inherent jitter present in the ground truth. On the other hand, it could be attributed to the dynamic optimization of accelerations performed by PIP.

Regarding the deployment performance, our method is significantly better than PIP. The time cost is reduced to 20% of that of PIP. Furthermore, our method obtains lower latency and higher frame rates on embedded computers. Since we no longer need future information, the latency is reduced to 12% of that of Transpose and TIP. Moreover, our method can still run at more than 60 fps on the embedded terminal, which allows for many potential application scenarios for our method. As shown in Table 2, our GPU usage rate is lower, which means that the device can allocate more computing resources for other functions. In addition, our occupancy has less float (only ±5%), and it can run more stably on the device.

We compared the distribution of the position and angular-related errors, as shown in Fig. 2. The joints error distribution (Fig. 2a) shows that 75 % of the errors of our method are less than 7.09 cm, which is better than Transpose's 8.14 cm and DIP's 9.75 cm. The mesh error distribution, as shown in Fig. 2b, shows that 75 % of the errors of our method are less than 7.46 cm, which is the best value among all models. From another two distribution plots (Fig. 2c, d), it is evident that FIP shows a peak value for the corresponding error closest to 0, and the peak is also higher. This indicates that FIP achieves angular errors concentrated within a narrower range compared to other methods, resulting in improved reconstruction outcomes. At the 75th percentile, FIP consistently performs the best. In terms of the *Aang* error, FIP outperforms the second-best method (PIP) by 11%, and in terms of the Ang error, FIP marginally surpasses PIP as well.

To qualitatively demonstrate the effectiveness of our method, we selected several actions to compare our method and other methods, as shown in Fig. 3. From (a) and (b), it can be observed that our result presents the most real reconstruction of the upper body, especially the angle between the arm and torso compared to other methods. Especially in (b), it can be seen that PIP reconstructs the pose with a slight forward tilt, and TIP fails to reconstruct the leg pose accurately, while

**Table 2 | Comparison of the deployment metrics on a GPU of the embedded computer**

| Method | MEM (Mb) | Usg (%) |
|---|---|---|
| Transpose | **732** | 60 (±10) |
| TIP | 910 | 50 (±8) |
| FIP(Ours) | 763 | **30** (±5) |

Bold denotes the best performance value for each metric.

our method is closest to the ground truth. Compared with Transpose, it can be seen from the side view of (d), (e), and (g) that our reconstruction of the upper body inclination is closer to the ground truth. The upper body reconstructed by Transpose always tends to lean forward, while ours is straighter. This is likely because we modeled the upper and lower bodies separately. Since Transpose models the body uniformly, the reconstruction of the upper body is more affected by the movements of the lower body. The lower body carries a high proportion of sensors but a low proportion of joints, so it may impact the reconstruction of the upper body, leading to a negative effect.

For more detailed comparison results, please refer to Supplementary Tab. 1 and Supplementary Tab. 2.

## Ablation study

We performed ablation experiments on the key submodules to show that the addition of our modules is effective, as shown in Table 3.

The most effective improvement is the human shape input. To verify the effect of inputting the human shape, we removed the skeleton input to the IK solver and half body regressors and retrained the pipeline ("no shape input" in Table 3), which causes an approximately 23% drop in the position accuracy. To meet the conditions of the kinematic inverse solution module based on just a single frame, we retrained a Multi-Layer perceptron (MLP) as a kinematic inverse solver for comparison ("MLP IK solver" in Table 3)[46,47]. The result shows that the adoption of the tree-like inverse solver reduces the average position error by 21%. To compare the effect of the shared model, we changed the integral model to input five sensors at the same time and output five position increments and retrained the pipeline ("no shared model" in Table 3). The results show that by sharing the same integral network for different sensors, not only is the number of parameters reduced, but the position reconstruction accuracy is improved by approximately 12%. As we added supervision on the SMPL skeleton parameters, to validate this setting, we compared it with the unsupervised case ("no skeleton supvision" in Table 3). The result shows that the supervision of the skeleton improves the position accuracy by an 8% and the SIP by 10%.

Finally, we set the shape parameters of SMPL to zero to obtain the average body parameters of the input (the height, arm length, and leg length) to compare the reconstruction effect of the body parameters on the model ("average shape" in Table 3). The results show that the measurements of the body parameters impact the reconstruction accuracy by approximately 5%.

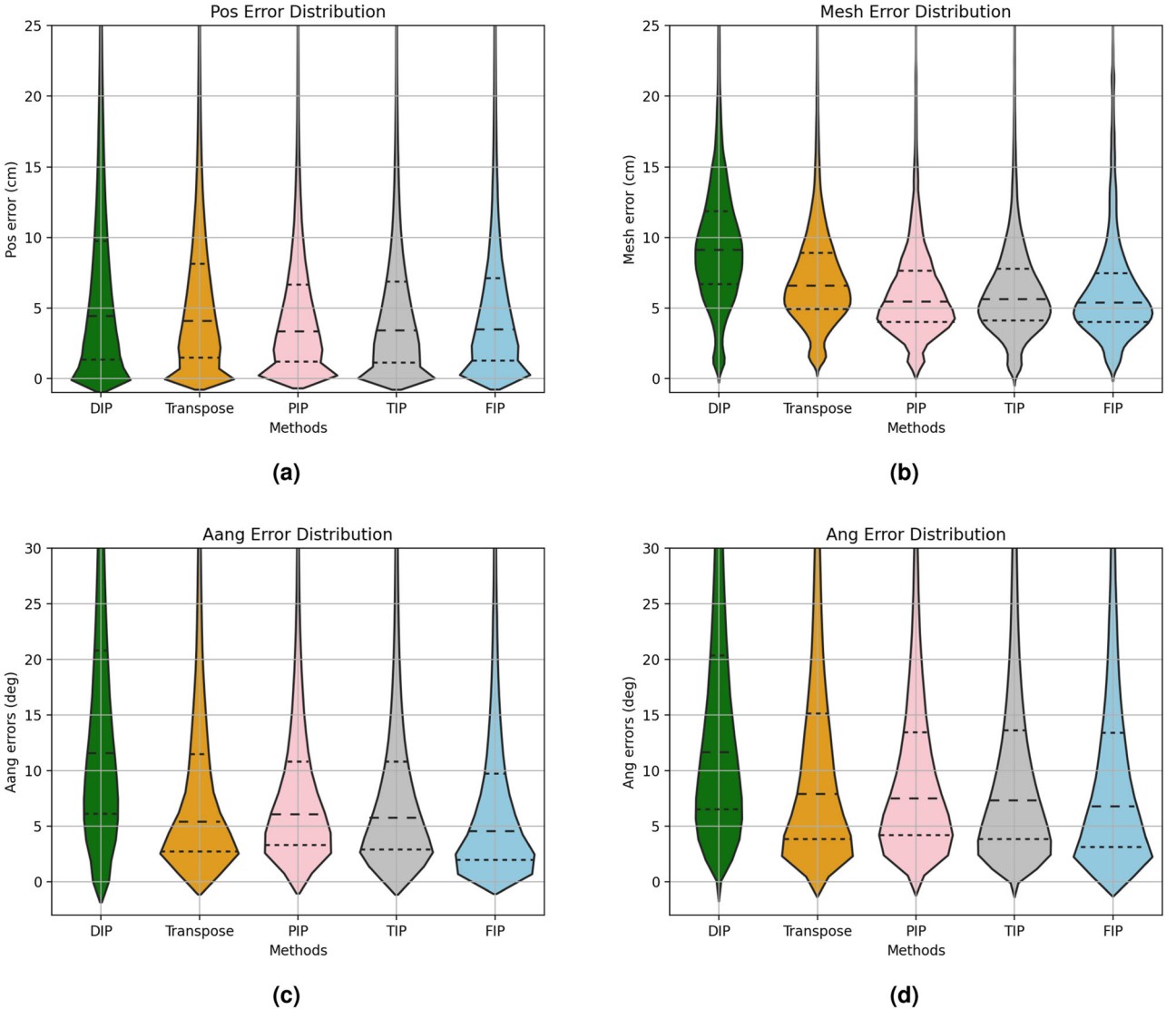

**Fig. 2 | Violin plots showing the errors of different methods.** In each violin, the three dashed lines from bottom to top represent the 25% percentile, 50% percentile, and 75% percentile, respectively. Panel (**a**) is the distribution of Position Error of the joints. The 75% percentile from left to right are 9.76cm, 8.14cm, 6.63cm, 6.85cm, and 7.09cm. Panel (**b**) represents the distribution of Mesh Error. The 75% percentile from left to right are 11.85cm, 8.91cm, 7.63cm, 7.76cm, and 7.46cm. Panel (**c**) denotes the distribution of Aang Error, where the 75% percentile from left to right are 20.80°, 11.49°, 10.80°, 10.83°, and 9.72°. And (**d**) is the distribution of Angle Error, where the 75% percentile from left to right are 20.37°, 15.16°, 13.44°, 13.62°, and 13.40°.

In summary, the addition of all modules is valid.

## Application

As shown in Fig. 4a, we built our live demo with Noitom sensors and the Unity platform. We presented the reconstruction of some motions (Fig. 4b). The live demo showcases the effectiveness of our method with regard to reconstructing human motions. Both our testing code and live demo are provided in the supplementary materials. For more action demonstrations, please refer to Supplementary Movie 1.

## Discussion

The experimental results show that our method significantly reduces the computation time per frame to 3 ms on a typical personal computer (PC), indicating a remarkable improvement in computing efficiency. This achievement enables the deployment of the model on mobile terminals with limited computational capabilities. As a result, the motion capture algorithm no longer requires a PC for calculation, and instead, the PC only needs to receive the calculated pose data. This advancement holds promising applications in real-time motion

capture scenarios involving multiple individuals. For instance, the models could be utilized to create digital twins of vast scenes or develop virtual multiplayer interactive games. The enhanced computing efficiency provides the possibility for the practical and widespread adoption of our method in various real-world applications.

However, this method still has some limitations. First, FIP relies on only six sensors, which prevents it from accurately estimating finger movements, wrist rotations, toe rotations and ankle rotations, despite their significance in motion analysis. To capture these actions in practical applications, additional devices such as data gloves are required. Second, the reconstruction results, as observed from quantitative metrics or demos, may not meet the demands for high precision in certain scenarios, such as joint medical analysis or skeleton animations. However, it is important to note that FIP still serves as a lightweight model that is suitable for a wide range of motion capture scenarios. Specifically, it can be applied in fields such as gaming and digital twinning. FIP's advantage lies in its ease of integration into wearable devices, such as VR/AR headset, enabling it to fulfill the motion capture needs of mobile platforms. Third, FIP lacks translation

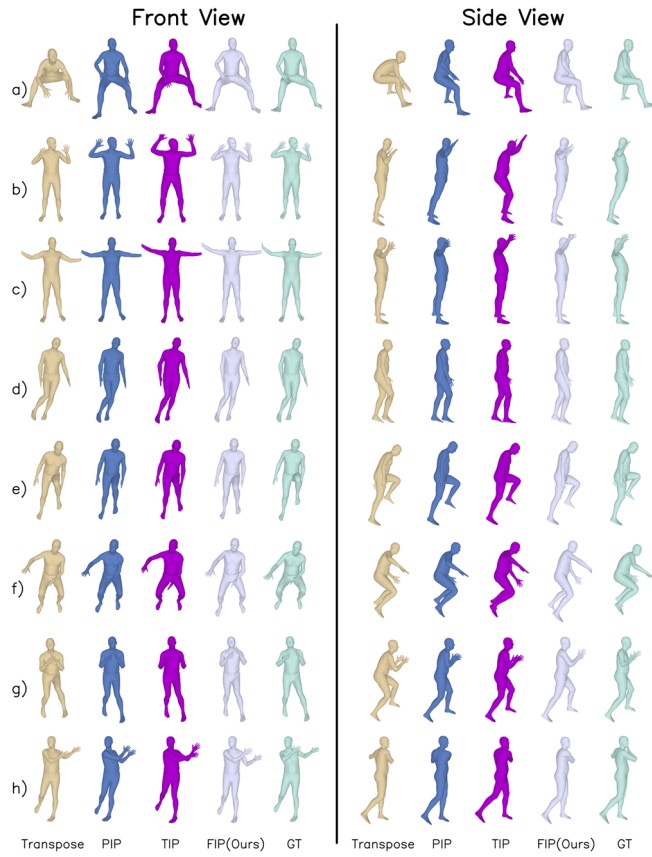

**Fig. 3 | Motion reconstruction of different methods.** Panels (**a**)–(**h**) represent some actions selected from the test set. The yellow color represents the reconstruction results obtained from Transpose, the blue color represents the reconstruction results obtained from Physical Inertial Poser (PIP), the purple color represents the reconstruction results obtained from Transformer Inertial Poser (TIP), the white color represents the reconstruction results obtained from Fast Inertial Poser (FIP), and the green color represents the Ground Truth (GT).

estimation compared to other similar methods like Transpose, PIP, and TIP, necessitating the use of alternative approaches to supplement translation information, such as human gait estimation or high-precision localization techniques like Simultaneous Localization and Mapping (SLAM).

In response to the aforementioned limitations, there are two primary directions for future work:

1. Enhancing the reconstruction accuracy: One approach is to continue improving the precision of the reconstruction by incorporating optimization designs similar to PIP or exploring alternative optimization strategies. Another possibility is to consider introducing additional types of sensors to capture more comprehensive information regarding joint movements or rotations. For example, using AR glasses aligned with the global coordinate system, and the global positions of other parts (such as the wrist) can be visually located.

2. Introducing root translation estimation: While we attempt to estimate a person's position using human gait in the video demonstration, it is important to note that this method is not inherently precise, and errors can accumulate over time. Therefore, it would be beneficial to explore the integration of more accurate positioning techniques such as SLAM, which can be integrated into VR/AR headsets to provide more reliable and absolute positioning information.

## Methods

We use 6 inertial sensors to estimate the motion of the human body in real time. The layout scheme is shown in Fig. 1, which is consistent

**Table 3 | Comparison of the improvements caused by each component**

|  | Method | SIP (deg) | Ang (deg) | Pos (cm) |
|---|---|---|---|---|
| Change of the different module | No shape input | 17.79 | 11.58 | 6.30 |
|  | MLP IK solver | 16.73 | 11.35 | 6.20 |
|  | No shared model | 16.05 | 10.61 | 5.71 |
|  | No skeleton supvision | 15.78 | 10.47 | 5.48 |
| Final fixed model | Average shape | 15.11 | 10.53 | 5.25 |
|  | FIP(Ours) | **14.37** | **10.06** | **5.09** |

Bold denotes the best performance value for each metric.

with the schemes of DIP[30], Transpose[35], PIP[37], and TIP[36]. The sensors are worn on the LAnkle, RAnkle, LWrist, RWrist, Head, and Pelvis nodes. We call the five nodes, except the pelvis node, leaf nodes below, while the pelvis is called the root node. Our method is based on the body coordinate system, as in Fig. 1. The origin is located at the root node, with the positive z-axis pointing forward, the positive x-axis pointing to the left, and the positive y-axis pointing upward. However, in the calculation process, we also use the global coordinate system, which does not move with the person's activity. Generally, its origin and axes coincide with the initial body coordinate system of the first frame.

The overall system pipeline is shown in Fig. 5. The input of the system is the 4 parameters (height, arm length, leg length, and sex) roughly measured from the human body and the normalized data of the IMUs (the data normalization process is explained in Section "Implementation details"), including the acceleration and rotation of the IMUs. The output is the rotation of the 15 joints excluding the ankles, wrists, and root node. The overall flow of Fast Inertial Poser (FIP) and our state-of-art kinematic inverse solver are introduced as follows.

### System input and output

We use the nodes of the SMPL model[42] to describe the joints of the body; therefore, *node* and *joint* are equivalent in the expression below. SMPL is currently a widely used human mesh model, based on which the open-source data are very abundant (such as the AMASS dataset[43]). We use the skeleton of the SMPL model as our human kinematics inference framework. The skeleton decomposes the whole body into 24 key nodes. It sets the root node at the pelvis, 4 nodes are used to represent a leg, 3 nodes are used to represent the spine, 2 nodes are used to represent a side of the chest and shoulders, 3 nodes are used to represent an arm, and 2 nodes are used to represent the neck through the head. Our human kinematics refer to the SMPL kinematics tree. The node numbers are shown in Fig. 6a.

We do not consider rotations of hands and feet. Therefore, in our study, we do not consider the rotations of Nodes 7, 8, 10, 11, 20, 21, 22, and 23. Moreover, the root node is provided by the sensor signal, so we only regress the rotation of the remaining 15 nodes.

**System input**. In the inference process, the inputs are body parameters $\mathbf{h}_p \in \mathbb{R}^4$. Considering the acceleration and rotation in the body coordinate system, acceleration vector and flattened rotation matrix from normalized IMU data are concatenated to obtain the related input $\mathbf{x}_0 \in \mathbb{R}^{12 \times J}$ ($J = 6$ here). During training, the first frame pose in the training segment $\mathbf{r}_i \in \mathbb{R}^{15 \times 9}$, which represents the initial pose, needs to be input into the model.

**System output**. During training, rotation of the 15 nodes and the positions of 19 nodes are output for supervision. During inference, positions are no longer output.

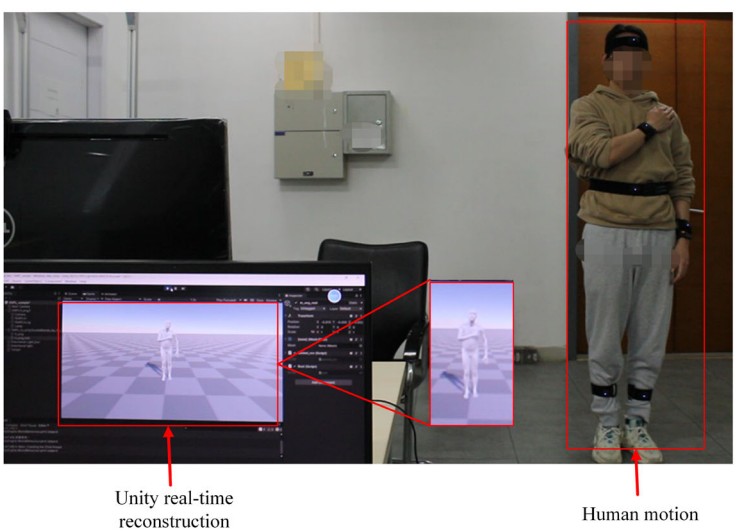

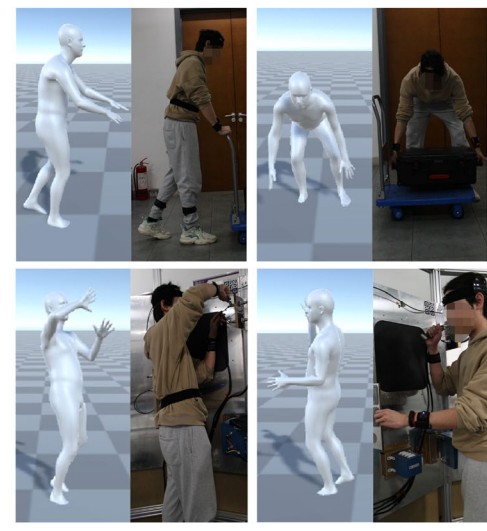

**(a)** Illustration of the live demo                                          **(b)** Reconstruction of real motion

**Fig. 4 | Application demo.** Panel (**a**) is an illustration of the live demo and we used the Noitom sensors and the Unity platform to reconstruct human motion in real time. Panel (**b**) is a reconstruction of real motion and we listed the reconstruction results of several common actions.

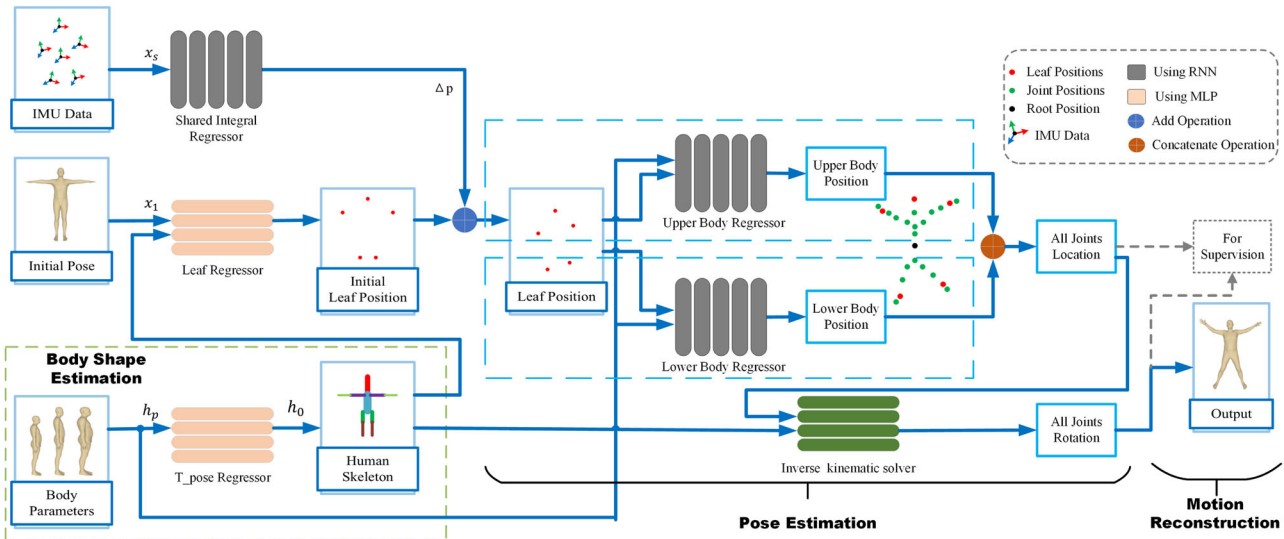

**Fig. 5 | Pipline of fast inertial poser.** There are three input items: (1) body measurement parameters; (2) initial pose; (3) IMU data. The body measurements are input into the T-pose Regressor to obtain the approximate human skeleton. The Leaf Regressor output the leaf node positions at the initial moment by the input of initial rotation and regressed skeleton. Shared Integral Regressor models the displacement increment of a leaf node by normalized IMU data. Then we add it with the initial leaf node position to obtain the leaf node position at the current moment. The leaf nodes are divided into upper body leaf nodes (wrists and head) and lower body leaf nodes (ankles). We perform separate upper body nodes regression and lower body nodes regression to obtain the node positions of the whole body. The node positions of the whole body and the approximate limb segment length of the human body are input into the Inverse Kinematic Solver to obtain rotation of the whole body nodes. The locations, rotations and skeleton are supervised during training.

## Key ideas

To design the structure, initially, we consider that the outputs of sensors are highly dependent on the human body shape, as shown in the factor graph in Fig. 7. $P$ represents the real pose of the human body, $\xi$ represents the observation, and $z$ represents the shape of the human body. The observations are jointly determined by the body shape and the real pose. Moreover, the body shape does not change over time, while the true pose changes.

When two people with very different body shapes perform the same movements, the distances moved by the sensors will be different due to the inconsistency in the segment dimensions of the two people.

Since the elapsed time is the same, it is easy to infer that the accelerometers will output different results. Therefore, it is necessary to take the human body shape information into consideration when using sparse IMU learning-based methods to estimate human poses. In addition, in contrast to the method of directly estimating the joint positions using sensor input, we use the method of estimating the leaf nodes' movement increment to estimate the positions of the leaf nodes at different times.

Generally, the proportions of most human bodies are similar. From a biological point of view, the sex is often the main factor that lead to differences in body proportions. Therefore, we include the sex

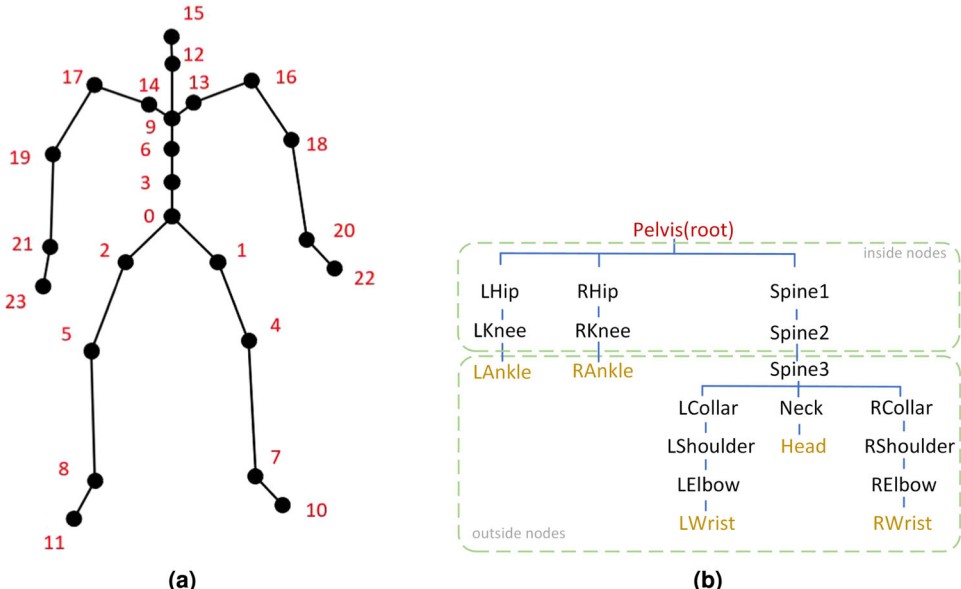

**(a)**                                        **(b)**

**Fig. 6 | The SMPL skeleton.** Panel (**a**) is the skeleton of the SMPL, which is extracted from the SMPL mesh model. Panel (**b**) is the human kinematic tree of SMPL. In (**b**), the color red represents the root node, while yellow represent the leaf nodes, which attach the sensors. We call the nodes 3 layers away from the root node *outside nodes*, and the nodes within 3 layers are called *inside nodes*.

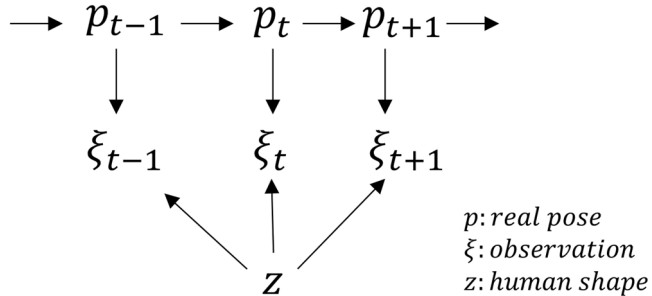

$p$ : *real pose*
$\xi$ : *observation*
$z$ : *human shape*

**Fig. 7 | Factor graph of the observation process.** *p* represents the real pose of the human body, *ξ* represents the observation, and *z* represents the shape of the human body. The observations are jointly determined by the body shape and the real pose, moreover the body shape does not change over time while the true pose changes.

as a necessary physical parameter. In addition to the sex, we select three other physical quantities that are easy to measure: height, arm length, and leg length, as shown in Fig. 1 (where arm length is measured from the wrist to the armpit, and leg length is from the ankle to the hip).

## Pipeline

Figure 5 shows the pipeline of FIP. We introduce the workflow of our model below. The RNN described below refers to the unidirectional RNN with LSTM cells[48].

The human body shape parameters are used to predict the human skeleton, which is involved in both the forward and inverse kinematic processes during movement. Therefore, when designing the network architecture, we predict the skeleton and incorporate it into relevant quantities. For example, after predicting the skeleton, we combine it with other information, such as inferring the initial state positions through the forward kinematic process and feeding them into the inverse solver.

Before introducing the details, Supplementary Fig. 1 shows the basic MLP module we use in the pipeline.

With the input of the body parameters, the T-pose regressor outputs the human skeleton $\mathbf{h}_0 \in \mathbb{R}^{57}$ (Supplementary Fig. 2).

We concatenated $\mathbf{h}_0$ and the initial rotation $\mathbf{r}_i$ to obtain $\mathbf{x}_1 \in \mathbb{R}^{192}$ and input it into an MLP to obtain the position of the leaf nodes at the initial moment.

Supplementary Fig. 3 shows the structure of the shared integral regressor. We used a shared weights RNN called the shared integral regressor, which regresses the leaf node positions at different frames. This regressor is used to simulate the displacement integration process with IMU noise. The input of the regressor is the data of a single sensor $\mathbf{x}_s \in \mathbb{R}^{12}$, which means that different sensors share the same integral regressor to obtain the displacement increment of the node. We added the increment with the initial position to obtain the leaf node positions $\mathbf{p}_{leaf}$.

Then, we use the RNN to regress all body node positions. We divide the whole body into the upper and lower body parts from the root node. The lower body only includes the 6 nodes of the legs, and the rest of the nodes belong to the upper body. Furthermore, since the kinematic chains of the upper body and the lower body are completely independent, we designed the regressors of the upper body and the lower body into two independent RNNs when designing the network architecture. We also divide the leaf node positions by upper body nodes (including wrists and head) and lower body nodes (ankles). Then, we encode the concatenation of the orientation of the sensors, positions of the upper/lower body nodes, and the human parameter-embedded vector, which is input into the corresponding RNN to obtain the positions of the half body nodes. For the upper and lower body, we use independent MLP encoders. In the experiment, we set the embedded vector's dimension to 128. Then, as shown in Supplementary Fig. 4 we merge the upper and lower body nodes to obtain the node positions of the whole body $\mathbf{p}_{all}$.

We input $\mathbf{p}_{all}$ and $\mathbf{h}_0$ into the kinematic inverse solver to obtain the rotation of the whole body nodes. During training, we supervise the skeleton, rotations and positions.

## Tree-like Inverse Kinematic Solver

The process of inverse kinematics in Euclidean space is non-differentiable. This means that if we use traditional inverse kinematics methods to supervise joint rotations, the gradients cannot propagate backward to the trained network parameters, making the model untrainable. It is necessary to use a differentiable model to

## BOX 1

# Submodule of the Inverse Kinematic Solver

**Input**: outside/inside nodes' positions $\mathbf{x}^{(0)}$, known nodes' orientation $\mathbf{r}_k$, outside/inside bones of predicted T-pose $\mathbf{b}_{sub}$

**Output**: orientation of outside/inside nodes.

1. $\mathbf{x}$ = concatenate[$\mathbf{h}_{sub}$, $\mathbf{x}^{(0)}$, $\mathbf{r}_k$]
2. $i$ = level(*leaf*)
3. $\mathbf{y}$ = empty queue
4. While $i >$ level(*devide*) do: {
    $\mathbf{r}_{i-1} = f^{(i)}(\mathbf{x})$
    $\mathbf{x}$ = concatenate[$\mathbf{x}$, $\mathbf{r}_{i-1}$]
    $i = i - 1$
    $\mathbf{y}$ = concatenate[$\mathbf{y}$, $\mathbf{r}_{i-1}$] }
5. **output y**

simulate the inverse kinematic process. Although there are such models ([46,47]), most of these models use a simple MLP for estimation, and they are not customized for this motion capture method. Therefore, they are not accurate enough, as shown by the ablation experiments.

Since the kinematic inverse solution is originally a static process, it is closer to physical reality that the model does not depend on time series. To make better use of the human motion tree information, we design a high-precision kinematics inverse solver. The structure is shown in Supplementary Fig. 5. In fact, the human kinematic and inverse process can be expressed by the following equations:

$$\mathbf{p}_j = \mathbf{p}_{j-1} + \left(\prod_{root}^{j-1} \mathbf{R}_i^{i+1}\right) \cdot \mathbf{b}_j \tag{1}$$

$$\mathbf{R}_j = f(\Omega(\mathbf{R}_{joint},\mathbf{p}_{joint}) \cup \{\mathbf{p}_{leaf},\mathbf{p}_j\}|parent = j, skeleton = \mathbf{h}) \tag{2}$$

We define a kinematic chain as a chain formed by nodes, which are traversed from the root node in a deep-first search manner to the leaf node. For example, (Pelvis-Spine1-Spine2-Spine3-Neck-Head) represents a kinematic chain passing through the neck in SMPL.

Eq. (1) represents the forward inference process of kinematics. $\mathbf{p}_j$ represents the position of the $j^{th}$ node; $\mathbf{R}_i^{i+1}$ represents the rotation from the $i^{th}$ node to the $(i+1)^{th}$ node on the chain; and $\mathbf{b}_j$ represents the relative position of $\mathbf{p}_{j-1}$ to $\mathbf{p}_j$ at Tpose. $\mathbf{h}$ is the skeleton parameter of the SMPL model, which is a $19 \times 3$ matrix composed of the relative positions of each node under T-pose, and $\mathbf{b}_j \in \mathbf{h}$. Eq. (2) represents the kinematic inverse process starting from a leaf node. $\mathbf{R}_j$ indicates the rotation of the $j^{th}$ node in the body coordinate system. *skeleton* represents the skeleton of the human body, which is composed of the relative positions of different nodes of the whole body. $\Omega\left(\mathbf{R}_{joint},\mathbf{p}_{joint}\right)|parent = j$ represents the position and rotation of the nodes after the $j^{th}$ node on a kinematic chain in the body coordinate system. $\mathbf{p}_{leaf}$ represents the position of the leaf node in the body coordinate system on the motion chain.

Taking the kinematic chain (Pelvis-Spine1-Spine2-Spine3-Neck-Head) as an example, before we begin to solve the rotation angle of node Spine1, through the reverse inference from the Head node, we obtain the rotation and positions of the Spine2, Spine3, and Neck nodes and the position of the Head node.

Actually, we can derive Eq. (2) from Eq. (1). In Eq. (3), $< \mathbf{p}_{j+1} - \mathbf{p}_j, \mathbf{b}_j >$ is the angle between the 2 vectors (which can be obtained by the cosine law). Mat$(\cdot)$ means converting the angle to a rotation matrix. In addition, $\mathbf{b}_j \in skeleton$. Thus, when

we progress down the chain toward the root node, we can find that $\mathbf{p}_j, \mathbf{R}_j$ are added to $\Omega(\mathbf{R}_{joint}, \mathbf{P}_{joint})$ one by one. Therefore, we can conclude that before calculating $\mathbf{R}_j$, information regarding the derivative joints by the $j^{th}$ joint (including the rotation and positions) can be obtained ($\Omega(\mathbf{R}_{joint}, \mathbf{P}_{joint})$). In addition, as Eq. (3), $\mathbf{P}_j = \mathbf{p}_j$ needs to be included. Thus far, it can be seen that it is reasonable to design our Inverse Kinematic (IK) solver by this equation.

$$\mathbf{R}_j = \prod_{root}^{j} \mathbf{R}_i^{i+1} = \text{Mat}(< \mathbf{p}_{j+1} - \mathbf{p}_j, \mathbf{b}_j >) \tag{3}$$

The design of our inverse solver refers to the human kinematic tree, which is shown in Fig. 6b. In the figure, red represents the root node, while yellow represents a leaf node, that is, the six nodes of the worn sensor. We call the nodes 3 layers away from the root node *outside nodes* and the nodes within 3 layers *inside nodes*. Since we do not consider the motion of hands and feet, we only consider the positions of the 19 nodes in the graph except the root node. We do not consider the rotation of the ankle and wrist, so we only consider the rotation angles of 15 nodes in total.

From Eq. (2), we further relax the condition. We consider that the nodes' information in the same layer on the kinematic tree is equivalent; *e.g.* the information carried by LCollor, Neck, and RCollar is equivalent. Our model of the process is shown in Eq. (4).

$$\bigcup_{j \in \text{level}(i)}\{\mathbf{R}_j\} = f(\Omega'(\mathbf{R}_j,\mathbf{P}_j|j \in \text{below}(i)) \cup \{\mathbf{P}_i,\mathbf{P}_{leaf}\}|\text{level}(i), skeleton = \mathbf{h}) \tag{4}$$

$\bigcup_{j \in \text{level}(i)}\{\mathbf{R}_j\}$ represents the set of rotation matrices of all nodes at the $i^{th}$ layer. $\Omega'(\mathbf{R}_j,\mathbf{P}_j|j \in \text{below}(i))$ represents the information of all nonleaf nodes below the $i^{th}$ layer (the leaf is below the root). $\mathbf{P}_i, \mathbf{P}_{leaf}$ represents the set of positions of all nodes at the $i^{th}$ layer along with the positions of all corresponding leaf nodes. Since leaf nodes are distributed at different levels in the motion tree, the nodes of interest are divided into two parts at level(3): outside nodes and inside nodes. We use the reverse order regression method to return from the bottom of the tree upward. When regressing the outside nodes, we only use the information of the leaf nodes of the upper body. We introduce the information of the other two leaf nodes of the lower body when regressing the inside nodes. For each layer, we use an MLP as the regressor.

Box 1 shows the algorithm flow of the submodule (including the inside nodes and outside nodes) of the Inverse Kinematic solver.

In the algorithm, level($\cdot$) represents a certain level on the kinematic tree. Regarding inside nodes, *level*(*leaf*) refers to the level corresponding to the ankle, while level(*divide*) corresponds to the layer in which the Pelvis is located, which is 0. Moreover, $r_k$ refers to the lower body IMU orientation and rotation of all outside nodes. Regarding outside nodes, level(*leaf*) refers to the level corresponding to the wrist, while level(*divide*) corresponds to the layer in which spine3 is located, which is 3. Additionally $r_k$ refers to the upper-body IMU orientation. $f^{(i)}$ is the regressor (MLP) of the $i^{th}$ layer. $r_i$ represents the rotation information of the $i^{th}$ layer node (utilizing 6D rotation continuous representation method). We use the concatenate operator to join new information to the set $\Omega'(\mathbf{R}_j,\mathbf{P}_j|j \in \text{below}(i))$.

## Ethical statement

This work is approved by the Science and Technology Ethics Committee, Tsinghua University.

The authors affirm that human research participants provided informed consent for the publication of the images in Figs. 1, 4.

In the demos and illustrations presented in this paper, the participants are two 24-year-old males, all of whom are authors of this paper. Informed consent has been obtained. Since the demos in this paper are for demonstration purposes only, gender and number of

participants are not considered. The remaining experimental results are based on datasets provided by open sources.

## Reporting summary

Further information on research design is available in the Nature Portfolio Reporting Summary linked to this article.

## Data availability

The test data and checkpoint generated in this study have been deposited in the Figshare database under accession code 10.6084/m9.figshare.25282732[49]. Source data are provided with this paper.

## Code availability

The test code of FIP can be accessed from github[50]: https://github.com/bachongyou/FIPinference.

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

## Acknowledgements
This work is supported by a grant from the Guoqiang Research Institute, Tsinghua University, China.(No. 2021GQG1018).

## Author contributions
Conceptualization: X.X, JF.Z; Methodology: X.X, JF.Z, JJ.W; Investigation: X.X, A.G, JJ.W, XY.Z; Experiment: X.X, A.G; Supervision: JF.Z, PF.F; Writing—original draft: X.X; Writing—review & editing: X.X, JF.Z, Funding acquisition and Project administration: JF.Z. All authors discussed the results and commented on the manuscript.

## Competing interests
The authors declare no competing interests.
