## [Peer Review File · Nature Communications]

Fast Human Motion reconstruction from sparse inertial measurement units considering the human shapeEditorial Note: Parts of this Peer Review File have been redacted as indicated to remove third-party material where no permission to publish could be obtained.

Reviewers' comments:

Reviewer #1 (Remarks to the Author):

The text concerns the construction of a neural network for the purpose of motion estimation using a sparse grid of 6 IMU sensors.

The text should definitely be edited in a way that makes it easier to read. Complete all details so that it is possible to train/test the method on other data. All abbreviations should be expanded and explained during the first appearance.

Motion acquisition is performed for biomechanical modeling and analysis. Unfortunately, this method is too inaccurate for that purpose. The second use is data for animation. Most often, these are skeletal animations. Please explain the need for the proposed method of data acquisition. Commentary on the possibility of using such estimation in skeletal animations with all limitations. The description of the method itself and the comparison to other methods should be described with more attention. In my opinion, this was not done reliably and certainly does not allow for drawing such strong conclusions - "The first real-time motion capture DNN approach that can be run on an embedded computer is proposed". This conclusion needs to be clarified as to what "real-time" and "embedded computer" mean. After all, other methods also managed to run (for comparison purposes) on an "embedded computer".

There are many parameters that are important when evaluating execution time and memory usage. Among other things, the libraries used, the version of the library, and the method of implementation. It is known that the code written earlier will be slower (uses older solutions) than the newer one. Therefore, the concept of computational complexity was introduced. Comparable runtime environments and software version impact should be considered when evaluating execution time. In addition, there is no information on whether comparable networks were trained on the same data set. What are the differences in GPU usage? Why PIP method does not use GPU power?

Orientation errors should be specified in a similar way to positions (like Figure 11). In order to evaluate the method, please provide errors for each segment/joint.

Errors should be described with a statistic (Table 2) - for example, min, max, mean, std.

In the main text, please include information about data statistics for training and testing data. What activities, how many samples...

Clarifications are needed for the video demo. It should be noted that the method generates data with sliding feet. There is also a leg-sinking phenomenon. This must be honestly described in the limitations. This is why this data is of very poor quality in skeletal animation.

The method does not allow to determine the translation. In the demo in the second part, there is a translation. How it was determined?

It should be clearly defined what data is collected from IMU sensors. The IMU sensor allows you to measure linear accelerations, magnetic field, and angular velocities. How the quality of the sensor (and also measurements) affects the method? What is the quality of the data used for training, testing, and demo?

Section 4.5 - The usability test should specify how the procedure was performed. What are the sensors, what values were taken, and what parameters (frequency and other settings)? In what coordinate system (sensor, global)? What additional preprocessing steps (for example calibration) were used?

More detailed remarks:

- Line 202 – define body coordinate system.
- Line 206 – define all parameters, also signals measured or estimated (How? What accuracy?) by IMU.
- Line 511 – define the global coordinate system.

Reviewer #2 (Remarks to the Author):

The proposed method seems to work well. The main improvement with respect to state of the art is the speed-up on more lightweight compute. From the video it seems that especially the motion of the arms might be better than prior work.

However, I have many concerns with (1) the writing and structuring of the paper, (2) motivation and clearer explanation of several parts and (3) the claimed speed up is actually a fundamental improvement with respect to state of the art.

For (1) I started making suggestions, they are not exhaustive. I feel first my other concerns should be addressed before I would revisit the writing and structuring of the paper.

For (2), an important example is the IK solver example, I don't see why this is different from doing IK to match proposed joint positions. It seems the algorithm works its way up from distal to proximal in the kinematic chain. Well, why is this better? What are downsides? Also, I think it could be formulated much clearer and the notation of equations 2 and 3 and algorithm 1 should be improved.

Next, the system has a modular setup where lower and upper body are split up. This seems helpful to reduce the sizes of the different regressors if I understand correctly? It is said in the paper that pose in upper and lower body are independent. This is not true, human motion is structured, and so the upper body pose gives a prior to what the lower body is doing. Such information is lost and could impact reconstruction quality.

Finally it should be motivated more clearly as to why the authors work in cartesian space to estimate leaf position, then go to joint rotation space to finally go back to cartesian space joint positions. Is this so that they stay "closer" to what IMUs measure (linear accelerations)? Why not directly predict joint rotations, the loss can still have the leaf positions. I realize this is a design choice, but it should be motivated because the made choice complicates the method significantly.

For (3), according to the PIP paper they have a latency of 16ms, so I do not understand how the authors here find 76ms. Second the TIP paper must at least have a latency of 83ms + inference time of the transformer, because they need 5 frames of future accelerations as input to their transformer. It thus seems not all results wrt prior work are reported accurately. Additionally concerning the contribution, FIP does not seem to estimate root translation estimation. This is not really touched upon in the paper and has been a hard problem that is at least partially tackled by PIP and TIP. Lacking this is a downside and should be discussed.

Remarks to writing and structure of the manuscript. And some questions.

I propose to call the system fast inertial poser rather than fast inertia poser to be in line with prior work.

The introduction is somewhat unstructured. For example the paragraph that starts on line 46 first brings up deploying a model on a mobile terminal and then shifts to the relation between body shape and sensor measurements. Structure one idea/shortcoming/... in a paragraph.

It is not clear whether FIP predicts root translation, this should be clear from the intro.

I feel the main shortcoming that is being addressed is the computational load of prior solutions.

This is discussed on the paragraph that starts on line 37. The next paragraph should then just introduce FIP and explain how it solves this problem. A subsequent paragraph can then discuss some technical details that explain why FIP can be faster while still being accurate.

Line 65: the sensor-shared integral RNN. This is not introduced before, so a reader has no clue what this means. This paragraph should be made much more clear on a higher level. This might take some more space.

Line 70: 'the kinematic inverse solution stage' \diamond reformulate: Our inverse kinematics solver takes joint positions and outputs joint rotations. Explain why this is a 'special' solver. For now it seems very standard IK to me...

The terms 'dense inertia' and 'sparse inertia' have no meaning to me. I understand what the

authors envision but it is better to use a more accurate description: f.e. a dense inertial sensor configuration.

Line 100: 'dense inertia' \diamond dense inertial sensor configurations

Line 120: Which other sensors?

Line 122-125: I don't see why skeleton calibration takes longer than for the sparse case. Because you need to measure the local position of the sensor on the bone? This needs to be more explicit..

Line 134: 'lack of joint constraints' \diamond this is not an accurate description. Reducing the number of sensors makes the IK problem that was determined with a dense configuration underdetermined.

Line 135: Just say that acceleration is noisy and therefore integration is problematic. 'The method needs to use the measurement as much as possible' is not a useful sentence here.

Line 204: What is the fourth parameter. Height, arm length, leg length, ... ?

Line 208: Why are ankles removed?

Line 228: Discarding foot rotation (i.e the ankle joint) seems spurious. This is an important joint when doing motion analysis.

Line 237: Specify the representation of orientation used (I assume flattened rotation matrix?)

Line 238-239: I do not understand, please explain better.

Line 254-255: This has not to do with the range of motion but with the segment dimensions

Figure 4: 'Obsavation' -- > observation

Reviewer #3 (Remarks to the Author):

Brief summary: The present work aimed at proposing a real-time motion capture approach with a new regression network architecture for the position of key joints with human shape input and a special inverse kinematic solver based on the human kinematic tree. Results demonstrate that the method is able to improve the inference speed and reduce the latency while ensuring the reconstruction accuracy.

Major comments:

The topic of this work is original, causing a probable interest in the reader. However, some changes are needed to improve the clarity of presentation and the comprehensibility of the study.

The abstract contains acronyms which are presented later in the text, but it is the first part to read. It would be advisable to define them also inside the abstract.

The introduction is well-written, even if the aim can be stated more clearly. Moreover, a brief description of IMUs with advantages and disadvantages especially related to optical systems is missing. Authors could consider these references to find useful details:

- Digo, E., Pastorelli, S., & Gastaldi, L. (2022). A Narrative Review on Wearable Inertial Sensors for Human Motion Tracking in Industrial Scenarios. *Robotics*, 11(6), 138.

- Filippeschi, A., Schmitz, N., Miezal, M., Bleser, G., Ruffaldi, E., & Stricker, D. (2017). Survey of motion tracking methods based on inertial sensors: A focus on upper limb human motion. *Sensors*, 17(6), 1257.

In addition, authors describe their model as divided in two stages, but in Figure 1 there are three blocks with three different colors (body shape estimation, pose estimation and motion reconstruction). I suggest clarifying this aspect in the text, better describing the process proposed in the figure. I would delete lines 84-85, because they represent a comment about results typical of the discussion section.

At the beginning of methods, authors should list where they positioned IMUs for a better clarity. Check the equations, because not all of them are cited in the text. I also suggest changing the title of subsection 3.2. because "Motivation" is not so clear.

Results are deeply and consistently discussed.

Minor comments:

- Line 3: I would define the motion capture as a technique instead of a technology
- Line 5: explain the acronyms VR and AR
- Line 71: explain the acronym SMPL
- Line 76: explain the acronym DNN
- Line 282: explain the acronym MLP

Dear Editor and Reviewers,

Thank you very much for your valuable comments and suggestions on our manuscript. We have revised and improved our manuscript according to your kind advices and detailed suggestions. The main revisions and corrections are highlighted in the manuscript, and the responses to the reviewers' comments are as follows. We sincerely hope this resubmission could be reconsidered.

Responses to reviewer 1

Reviewer Comments:

The text concerns the construction of a neural network for the purpose of motion estimation using a sparse grid of 6 IMU sensors.

(1) **Comment:** The text should definitely be edited in a way that makes it easier to read. Complete all details so that it is possible to train/test the method on other data.

Response: Thanks a lot for all your valuable comments. We have completely restructured the manuscript based on journal template to make it easier to read, and uploaded all source code and supplements.

(2) **Comment:** All abbreviations should be expanded and explained during the first appearance.

Response: In the revised version of the manuscript, we have addressed this concern by modifying the first occurrence of certain terms, such as VR, AR (page1 line 9), DNN (page2 line 19), etc.

(3) **Comment:** Motion acquisition is performed for biomechanical modeling and analysis. Unfortunately, this method is too inaccurate for that purpose. The second use is data for animation. Most often, these are skeletal animations. Please explain the need for the proposed method of data acquisition. Commentary on the possibility of using such estimation in skeletal animations with all limitations.

Response: Unfortunately, it is true that our method does have certain limitations in certain scenarios. In the revised version, we have provided a more detailed discussion on this limitation (page 8 lines 4-8). In summary, the accuracy of the method may not meet the requirements of high-precision applications in fields such as biomechanical modeling and analysis.

However, due to its high computational efficiency, our method is well-suited for integration into wearable devices, making it suitable for a wide range of other applications. For example, in the gaming industry using VR and AR technologies, our method can be directly integrated into VR or AR headsets with limited computational power. Additionally, it is also applicable to large-scale motion capture scenarios such as digital twins in production workshops. (Supplementary discussion in the text page 8 lines 8-11)

(4) **Comment:** The description of the method itself and the comparison to other methods should be described with more attention. In my opinion, this was not done reliably and certainly does not allow for drawing such strong conclusions - "The first real-time motion capture DNN approach that can be run on an embedded computer is proposed". This conclusion needs to be

clarified as to what "real-time" and "embedded computer" mean. After all, other methods also managed to run (for comparison purposes) on an "embedded computer".

Response: We have carefully revised the manuscript based on the subsequent feedback to improve the description of the proposed method and provide more comprehensive details on the comparison with other methods. As corrected in page3 lines 28 of the revised manuscript: "A real-time (more than 60 FPS) motion capture DNN approach that can be run on an embedded computer is proposed." The term "embedded computer" refers to traditional miniature computers such as Raspberry Pi or Nvidia's Jetson series onboard computers. These computers are characterized by limited computing power but small form factor, for example, the Jetson TX2 has dimensions of only 50mm*87mm, making it suitable for integration into wearable devices or small-scale equipment. The term "real-time" mentioned here refers to a frame rate exceeding 60 FPS and latency below 15ms. In the original text, for comparative purposes, other methods were also evaluated on embedded computers (as shown in Tables 2 and 3), but these methods did not meet the same performance criteria. Regarding the description of embedded computers, we have made a supplement in page3 lines 5-6 of the revision.

- (5) **Comment:** There are many parameters that are important when evaluating execution time and memory usage. Among other things, the libraries used, the version of the library, and the method of implementation. It is known that the code written earlier will be slower (uses older solutions) than the newer one. Therefore, the concept of computational complexity was introduced. Comparable runtime environments and software version impact should be considered when evaluating execution time. In addition, there is no information on whether comparable networks were trained on the same data set. What are the differences in GPU usage? Why PIP method does not use GPU power?

Response: In the revised version, in page 15 lines 17-19 "Model training", we specified the training conditions. And in page4 lines 6-7, we specified the comparison conditions. For the comparative analysis, all methods were evaluated using the same device and code environment, namely PyTorch 1.8 and CUDA 11.2 on a graphics workstation, and Jetpack PyTorch 1.8 and CUDA 10.2 on the TX2NX (as described in the initial manuscript). Furthermore, the comparison models used in the study were obtained from the official parameters provided by the authors of the respective papers¹⁻⁴ (references mentioned in the original text). These papers also detail the training procedures they employed. To ensure consistency in the comparison, we pretrained all models on the AMASS dataset and fine-tuned them using data from the first eight participants of the DIP-IMU dataset.

Furthermore, since several of these methods are based on deep neural networks, utilizing GPUs for deep neural network computations can significantly accelerate the overall computational process. This acceleration is evident in DIP, Transpose, TIP, and FIP. However, it should be noted that in the case of PIP, a substantial portion of the computational workload is involved in subsequent optimizations, which primarily rely on CPU computations. As stated in the PIP source code (<https://github.com/Xinyu-Yi/PIP/blob/main/readme.md> under "Install dependencies"), the inference part of the model is intended to be executed on the CPU. Consequently, for the purpose of comparison, the PIP method was not executed on a GPU.

- (6) **Comment:** Orientation errors should be specified in a similar way to positions (like Figure 11).

In order to evaluate the method, please provide errors for each segment/joint. Errors should be described with a statistic (Table 2) - for example, min, max, mean, std

Response: Following the revisions, we have added error distribution plots related to angle in similar way to positions in Fig.2 c), d).

And errors for each joint rotations with a statistic (mean, std) are also provided in Appendix B, titled "More Results".

	DIP	Tanspose	PIP	TIP	FIP
Lhip	12.91 (+/- 10.25)	10.98 (+/- 9.48)	9.79 (+/- 8.10)	9.42 (+/- 10.34)	8.91 (+/- 6.11)
Rhip	12.14 (+/- 10.11)	10.58 (+/- 9.07)	9.27 (+/- 6.86)	8.71 (+/- 9.89)	8.61 (+/- 5.69)
Spine1	8.04 (+/- 5.87)	9.29 (+/- 6.69)	7.50 (+/- 5.02)	7.78 (+/- 5.55)	7.99 (+/- 5.51)
Lknee	10.23 (+/- 6.35)	3.60 (+/- 2.46)	4.75 (+/- 3.28)	3.85 (+/- 2.57)	3.29 (+/- 3.30)
Rknee	9.35 (+/- 6.15)	3.63 (+/- 2.81)	4.82 (+/- 3.58)	3.70 (+/- 3.01)	3.60 (+/- 3.52)
Spine2	10.48 (+/- 7.61)	11.88 (+/- 7.32)	9.39 (+/- 6.25)	10.17 (+/- 6.51)	10.62 (+/- 7.14)
Spine3	12.58 (+/- 9.38)	13.33 (+/- 8.19)	10.90 (+/- 7.45)	12.16 (+/- 7.86)	13.01 (+/- 8.75)
Neck	8.71 (+/- 5.52)	7.27 (+/- 4.59)	7.75 (+/- 5.02)	7.06 (+/- 4.45)	7.49 (+/- 4.89)
Lcollar	18.46 (+/- 11.63)	19.61 (+/- 11.43)	16.36 (+/- 10.78)	17.70 (+/- 11.49)	18.49 (+/- 11.60)
Rcollar	18.29 (+/- 10.71)	17.46 (+/- 10.06)	16.81 (+/- 11.15)	16.81 (+/- 10.67)	17.19 (+/- 10.29)
Head	9.61 (+/- 5.95)	2.88 (+/- 1.95)	4.57 (+/- 3.32)	3.71 (+/- 2.32)	2.74 (+/- 3.05)
Lshoulder	23.72 (+/- 15.86)	23.91 (+/- 14.77)	20.64 (+/- 14.07)	23.04 (+/- 15.66)	21.66 (+/- 15.15)
Rshouder	25.19 (+/- 15.31)	22.27 (+/- 12.93)	20.65 (+/- 14.69)	20.82 (+/- 15.19)	18.78 (+/- 13.99)
Lelbow	26.59 (+/- 19.79)	6.50 (+/- 4.84)	7.33 (+/- 5.46)	8.39 (+/- 6.77)	4.39 (+/- 4.22)
Relbow	27.65 (+/- 21.23)	6.12 (+/- 5.06)	7.12 (+/- 5.77)	7.82 (+/- 6.91)	4.45 (+/- 4.40)

Table 1. Comparison of the mean value and standard deviation of the angle errors for all nodes.

It can be seen that our method exhibits better performance in terms of angle-related error distribution, characterized by smaller mean values and tighter distributions, as indicated by the peak being closer to zero and having a higher peak value.

Since the errors related to Tab. 1 have been represented in the error distribution chart shown in Fig. 2, the statistical characteristics of errors can be more directly observed. Therefore, additional statistical metrics are not added to the table due to space constraints.

(7) **Comment:** In the main text, please include information about data statistics for training and testing data. What activities, how many samples...

Response: In the main text, we have cited and referenced these datasets, including the AMASS dataset and the DIP-IMU dataset^{1,5}. Moreover, we have provided a brief introduction to these datasets in Appendix Section A.1 and Page3 lines 39-40. The AMASS dataset consists of over 10,000 motions, while the DIP-IMU dataset contains more than 50 motions.

(8) **Comment:** Clarifications are needed for the video demo. It should be noted that the method generates data with sliding feet. There is also a leg-sinking phenomenon. This must be honestly described in the limitations. This is why this data is of very poor quality in skeletal animation.

Response: We have made additions to address this concern in Section Discussion to discuss the limitation and future work (page 8 lines 4-11). As mentioned in the third point of your Comment, we acknowledge that our method does exhibit a certain degree of accuracy issues.

(9) **Comment:** The method does not allow to determine the translation. In the demo in the second part, there is a translation. How it was determined?

Response: In the demo, we employed a gait-based position estimation method, and the specific implementation details are provided in Appendix Section A.5.2. The whole section is to illustrate how FIP is integrated with gait-based position estimation methods. We have included a thorough derivation and explanation of the position estimation approach used in the demo to clarify how the position estimation is performed.

(10) **Comment:** It should be clearly defined what data is collected from IMU sensors. The IMU sensor allows you to measure linear accelerations, magnetic field, and angular velocities. How the quality of the sensor (and also measurements) affects the method? What is the quality of the data used for training, testing, and demo?

Response: As you correctly mentioned, the 9-axis sensor can output linear acceleration, angular velocity, and magnetic field strength. In our case, the linear acceleration is directly output by the sensor itself, while the orientation is obtained through filtering and fusion of the raw signals using internal algorithms of the IMU, and is output in the form of a rotation matrix or quaternion. This clarification has been added in Appendix Section A.4 after the revision. In the revision, we also provided an explanation, referring to page 8 line 46.

However, it is challenging to assess the impact of sensor quality on the reconstruction, mainly due to the lack of information about the sensor models or the accuracy of the data in the dataset. Nevertheless, through certain experiments, it can be determined that the model is data-driven, and hence it exhibits a certain level of robustness to the accuracy of the sensors. It can achieve satisfactory results without the need for extremely strict calibration. As for the impact of body parameter measurements on the reconstruction results, the ablation study in the paper compares the "average parameters" and "specific parameters" (Tab. 3). The results indicate that there is a certain level of error, but the average error of joint positions is within 0.5 cm.

For the quality of data used for training and testing, the attributes of the training and testing data can be found in the references of their respective datasets^{1,5}. In Appendix Section A.5.1, we have provided information about the specific IMU model⁶ used in the demo to show the data quality of demo.

(11) **Comment:** Section 4.5 - The usability test should specify how the procedure was performed. What are the sensors, what values were taken, and what parameters (frequency and other

settings)? In what coordinate system (sensor, global)? What additional preprocessing steps (for example calibration) were used?

Response: As mentioned in comment (10), an introduction to the IMU is provided in Appendix A.4, and detailed information about the IMU model used in the demo and its corresponding coordinate system is explained in Appendix A.5.1. The IMU used in the demo is the Noitom PN3 (60fps, 0.02° angular resolution), and the raw output data is computed based on the ENU (East North Up) coordinate system.

The calibration of the ENU coordinate system and the global coordinate system is performed using the first frame at the start of the demo. The specific preprocessing steps are described and explained in detail in Appendix A.5.1.

(12)Comment: More detailed remarks:

- Line 202 – define body coordinate system.
- Line 206 – define all parameters, also signals measured or estimated (How? What accuracy?) by IMU.
- Line 511 – define the global coordinate system.

Response: The following is a supplemental modification addressing these details:

- 1) The definition of the human body coordinate system is described in Figure 1 of the main text. Furthermore, it has been supplemented and clarified in Appendix A.6 as follows: "The human body coordinate system is defined according to the SMPL coordinate system. In the T-pose position, the x-axis points towards the left hand direction, the y-axis points upwards, and the z-axis points towards the front of the body. This coordinate system follows the standard conventions used in human pose estimation."
- 2) The parameters, including height, arm length, leg length, and sex, have been described. As for the IMU-related concerns mentioned earlier, the measurement method and accuracy of the IMU are discussed in detail in Appendices A.4 and A.5.1. Regarding the training and testing data, it should be noted that the IMU data has already been converted to the global coordinate system by the dataset, and thus no additional preprocessing is performed.
- 3) The definition of the global coordinate system is provided in Appendix A.6 and supplemented in page 8 lines 27-29 of the revision. The global coordinate system is fixed and does not move with the motion of the human. During the experiment, the initial position and orientation of the human body coordinate system are calibrated to align with the global coordinate system.

Responses to reviewer 2

Reviewer Comments: The proposed method seems to work well. The main improvement with respect to state of the art is the speed-up on more lightweight compute. From the video it seems that especially the motion of the arms might be better than prior work.

However, I have many concerns with (1) the writing and structuring of the paper, (2) motivation and clearer explanation of several parts and (3) the claimed speed up is actually a fundamental improvement with respect to state of the art.

(1) **Comment:** For (1) I started making suggestions, they are not exhaustive. I feel first my other concerns should be addressed before I would revisit the writing and structuring of the paper.

Response: Thanks a lot for all your valuable comments. We have restructured the manuscript and improved the text to make it easier to read.

(2) **Comment:** For (2), an important example is the IK solver example, I don't see why this is different from doing IK to match proposed joint positions. It seems the algorithm works its way up from distal to proximal in the kinematic chain. Well, why is this better? What are downsides? Also, I think it could be formulated much clearer and the notation of equations 2 and 3 and algorithm 1 should be improved.

Response: In the beginning of revision section "Tree-like Inverse Kinematic Solver" (page 11 line 4), it is mentioned, "Since the real kinematic inverse solution process is not differentiable, the overall model cannot be effectively trained."

In neural network-based methods, the derivatives of the loss function with respect to the parameters are computed to perform gradient descent and effectively train the neural network parameters. However, in the real kinematic inverse process, i.e., Equation (3) in the revision, solving for the angle between two vectors is not a differentiable process. This is primarily because the three-dimensional representation of rotations in space is not continuous (e.g., 360° and 0° are actually the same angle, leading to ambiguity), making it impossible to compute the derivatives of the loss function with respect to the parameters. This prevents the neural network parameters of the method from being trained. On the other hand, the forward kinematic derivation process is continuously differentiable. Therefore, we designed an inverse solver model to approximate the non-differentiable kinematic inverse process and supervised it using the differentiable forward kinematic derivation process. As mentioned in the comment, we aim to make the designed kinematic inverse solver closer to the real kinematic inverse process. Additionally, since the sensors are deployed at the end of the human body, the information from the extremities of the body is expected to be closer to the real-world situation. Therefore, we adopted a solution approach from the limb ends towards the root.

In the revised version, we have added explanations for the symbols \mathbf{h} and \vec{b}_j of Eq.2 (page 11 lines 16- 18) and provided explanations for the variable r_i in Algorithm 1 (page 14 lines 1-3). In addition, we also modified the description of Eq.4 in page 12 to make it different from the conventional IK process, and unified the symbols in other equations.

(3) **Comment:** Next, the system has a modular setup where lower and upper body are split up. This seems helpful to reduce the sizes of the different regressors if I understand correctly? It is said in the paper that pose in upper and lower body are independent. This is not true, human motion is structured, and so the upper body pose gives a prior to what the lower body is doing. Such information is lost and could impact reconstruction quality.

Response: After the revision, in Appendix C.1 "Explanations of the network structure," we provide further clarification. It is correct that there is coupling between the upper and lower body movements. The "independent" mentioned here refers to the regression of the upper and lower body separately in the network structure. As described in the added explanation: "However, due to the coupling between the upper and lower body movements, independent

regression models do not completely decouple this coupling. This is because, firstly, during the estimation of node positions, the estimated skeleton is divided into upper and lower body inputs, which are generated by a neural network, thus introducing a certain level of coupling. Secondly, it is a kind of data-driven method, and during the training process, there may exist 'latent' connections between the parameters of the two independent regressors. Lastly, the method does not fully decouple the estimation of the upper and lower body, as they are still unified in the kinematic inverse process."

Indeed, the lower body has only 2 sensor nodes, while it only needs to estimate the positions of 4 nodes. On the other hand, the upper body has 3 sensor nodes but needs to estimate the positions of 11 nodes. If the same model is used to estimate the positions of all nodes, it is likely that the lower body would have a "disproportionately large" impact on the overall estimation. As shown in the additional experimental results in Appendix B "More Results," this design can indeed improve the accuracy of rotation estimation for each node to some extent.

- (4) **Comment:** Finally it should be motivated more clearly as to why the authors work in cartesian space to estimate leaf position, then go to joint rotation space to finally go back to cartesian space joint positions. Is this so that they stay "closer" to what IMUs measure (linear accelerations)? Why not directly predict joint rotations, the loss can still have the leaf positions. I realize this is a design choice, but it should be motivated because the made choice complicates the method significantly.

Response: The clarifications and explanations:

Firstly, in the usage of the model, there is no final step of returning to the Cartesian joint coordinate system. In the training process, it is true that we return to the Cartesian space to supervise the training of the kinematic inverse solver. However, in practical usage, we only obtain the rotation angles of each joint and apply the SMPL model for reconstruction, without further returning to the Cartesian space.

As for why we choose to first predict joint positions and then predict joint angles, the main reasons are as follows:

- 1) Directly predicting joint angles would partially lose the acceleration information from the IMU. Although using a single IMU to estimate the object's position is not highly accurate, it still provides relevant information to some extent. If we only constrain the position in the loss function, the true physical characteristics of the IMU would be compromised. Directly using predicted joint angles to infer joint positions would not control the trained parameters to reflect the true spatial information of the IMU since it is equivalent to only estimating the limb orientation in the inference process. On the other hand, the IMU's orientation is highly accurate and can directly describe joint angles. Therefore, even if joint angles are estimated in the second stage, the IMU's orientation information is not compromised. Conversely, the acceleration information from the IMU cannot directly reflect joint positions. Hence, it is necessary to first characterize the positions of the nodes that the IMU is attached to in space, i.e., estimating the positions of the human body nodes in the Cartesian coordinate system. This theoretically improves the model's representational capacity.
- 2) In our method, we first estimate the skeleton parameters of the human body, which are the

positions of each joint in the Cartesian coordinate system under the T-pose. By first estimating the positions of the joints in the Cartesian coordinate system and then obtaining the joint angles through kinematic inverse solving, we can combine the model with the skeleton parameters in the form of kinematic inverse solving. This approach allows us to supervise the skeleton estimator with real values and supervise both the kinematic inverse solver and the skeleton estimator with kinematic forward inference ground truth values. Therefore, this approach of first estimating the positions of the human body joints in the Cartesian coordinate system and then estimating the joint angles can effectively incorporate human body parameters.

- 3) In previous methods, many have also followed the practice of first estimating Cartesian coordinate positions and then estimating joint angles, such as DIP¹, Transpose², and PIP³. It is reasonable to borrow from these methods.

- (5) **Comment:** For (3), according to the PIP paper they have a latency of 16ms, so I do not understand how the authors here find 76ms. Second the TIP paper must at least have a latency of 83ms + inference time of the transformer, because they need 5 frames of future accelerations as input to their transformer. It thus seems not all results wrt prior work are reported accurately.

Response: Regarding this feedback, we consider it to be an unfounded accusation. After reviewing the papers and source code of the previous work upon receiving the feedback, we can ensure that the experimental results presented in our paper are entirely accurate. As for the mentioned misunderstanding in the feedback, we would like to provide an explanation and clarification here.

1. In the PIP and Transpose papers, their experimental conditions are different from ours. In order to compare the performance of the methods on embedded devices, we used TX2NX to validate the performance in our experimental comparisons, as mentioned in the original manuscript's section 4.1 "Implementation details". On the other hand, their experiments were conducted on a laptop with an Intel i7 processor and a RTX2080 graphic card, which has much higher computational performance than the embedded device TX2NX. This difference in computational capabilities may lead to shorter latency and computation times in their experiments. Therefore, these differences do not conflict with our experimental results. In other words, using a more efficient computing platform can reduce the latency of PIP to 16ms, but on a limited computing platform like TX2NX, it indeed takes 76ms. In addition, we provide the test source code, and when running our solution on the PC with a RTX2060 graphic card, it is found that our method has a latency of only about 3ms.

2. The TIP method utilizes a transformer. After examining their open-source code and paper, we found that it does not use information from future frames. In contrast, the DIP and Transpose method incorporates information from the future 5 frames. Therefore, we suspect that the reviewer may have confused these methods.

- (6) **Comment:** Additionally concerning the contribution, FIP does not seem to estimate root translation estimation. This is not really touched upon in the paper and has been a hard problem that is at least partially tackled by PIP and TIP. Lacking this is a downside and should be

discussed.

Response: In fact, this point is addressed in Section 5 "Conclusion and Future Work" of the original paper, and in section Discussion of the revision (page 8 lines 11-13). FIP itself is a pose estimation algorithm and indeed cannot directly estimate the position of the human body. However, this does not hinder its combination with other methods that estimate the position, such as high-precision SLAM methods or the location estimation algorithms mentioned in the comment, such as TIP and PIP. In our demo, we attempted to estimate the position of the human body using a gait-based method, as mentioned in the revised Appendix A.5.2 "Translation estimation". However, since this was not the main focus or novelty of our paper (unlike TIP).

Remarks to writing and structure of the manuscript. And some questions.

- (7) **Comment:** I propose to call the system fast inertial poser rather than fast inertia poser to be in line with prior work.

Response: Thank you for your feedback. We have made revisions as suggested.

- (8) **Comment:** The introduction is somewhat unstructured. For example the paragraph that starts on line 46 first brings up deploying a model on a mobile terminal and then shifts to the relation between body shape and sensor measurements. Structure one idea/shortcoming/... in a paragraph.

Response: In the revised version, we have removed unnecessary statements to make it more concise: "This method focuses on obtaining faster performance by introducing the physical parameters of the human body shape, while the previous method did not consider this" (page 3, lines 14-15). In addition, we have reorganized the introduction to address these issues.

- (9) **Comment:** It is not clear whether FIP predicts root translation, this should be clear from the intro.

Response: This comment is consistent with the meaning expressed in Comment 6. Please refer to the response to Comment 6. Human pose estimation and position estimation are two separate research directions. Human pose estimation focuses on reconstructing the positions or rotations of individual joints in the human body, while position estimation focuses on determining the spatial location of the human in the environment. Although these two directions can potentially be integrated, this paper primarily addresses the reconstruction of individual joints in the human body.

- (10) **Comment:** I feel the main shortcoming that is being addressed is the computational load of prior solutions. This is discussed on the paragraph that starts on line 37. The next paragraph should then just introduce FIP and explain how it solves this problem. A subsequent paragraph can then discuss some technical details that explain why FIP can be faster while still being accurate.

Response: In the revised version, adjustments have been made to the description in accordance with the comment at page 3 lines 15-18. And Appendix C.2 "Analysis of

deployment improvement" has been added, which provides an analysis of the computational efficiency improvements. The analysis mainly focuses on three points: first, the cancellation of optimization designs; second, the ability of this method's network structure to achieve stronger expressive power with a smaller model; and third, the cancellation of the bidirectional propagation mechanism in the RNN.

(11)Comment: Line 65: the sensor-shared integral RNN. This is not introduced before, so a reader has no clue what this means. This paragraph should be made much more clear on a higher level. This might take some more space.

Response: In the revised version, we have added an explanation to clarify this point: "In other words, different sensors share the same integral RNN to estimate their displacements" in page 3 lines 21-22.

(12)Comment: Line 70: 'the kinematic inverse solution stage' reformulate: Our inverse kinematics solver takes joint positions and outputs joint rotations. Explain why this is a 'special' solver. For now it seems very standard IK to me...

Response: This comment is consistent with the sentiment expressed in Comment 2. Please refer to the response provided in Comment 1 for further clarification. Meanwhile, we have modified the description of Equation 4 to make it appear different from the traditional IK process.

(13)Comment: The terms 'dense inertia' and 'sparse inertia' have no meaning to me. I understand what the authors envision but it is better to use a more accurate description: f.e. a dense inertial sensor configuration. Line 100: 'dense inertia' dense inertial sensor configurations

Response: We have replaced the relevant description, as shown in page 1 lines 23-24, 27 and 36, to make the description clearer. We have added the explanation of the dense inertial configuration in the revision (page 1 lines 24-26), and added a reference in the revised version where readers can access the official website for further information (like the figure below).

[REDACTED]

- (14)Comment:** Line 120: Which other sensors?
Response: In the cited article, it is mentioned that Ultra-Wideband (UWB) sensor is used. In the revised version, we have provided additional clarification on this matter(page 1 line 33)
- (15)Comment:** Line 122-125: I don't see why skeleton calibration takes longer than for the sparse case. Because you need to measure the local position of the sensor on the bone? This needs to be more explicit..
Response: In practice, when using these solutions, certain actions need to be performed to calibrate the transformation relationship between the skeleton and the sensors. Additionally, due to the larger number of sensors, there is a higher likelihood of loosening, which often prolongs the calibration process. During our own experiments with dense sensors, this process typically takes around twenty minutes. However, sparse solutions typically only require wearing the sensors in the specified orientations and do not heavily rely on the body's own skeleton. As a result, this process does not require much time and can be completed relatively quickly.
- (16)Comment:** Line 134: 'lack of joint constraints' this is not an accurate description. Reducing the number of sensors makes the IK problem that was determined with a dense configuration underdetermined
Response: The corresponding section has been revised accordingly (Page 2 lines 7-8).
- (17)Comment:** Line 135: Just say that acceleration is noisy and therefore integration is problematic. 'The method needs to use the measurement as much as possible' is not a useful sentence here
Response: In the revised version, we have made modifications to the relevant sentence in page 2 lines 8-9.
- (18)Comment:** Line 204: What is the fourth parameter. Height, arm length, leg length, ... ?
Response: It is sex. In the revised version, we have highlighted this point (page 8 lines 30-31) to emphasize its significance.
- (19)Comment:** Line 208: Why are ankles removed? Line 228: Discarding foot rotation (i.e the ankle joint) seems spurious. This is an important joint when doing motion analysis.
Response: We have added the limitation discussion in the section Discussion (Page 8, lines 4-8), to discuss the reasons behind these limitations and potential solutions.
- (20)Comment:** Line 237: Specify the representation of orientation used (I assume flattened rotation matrix?)
Response: Yes, that's correct. It is indeed a flattened rotation matrix. In the revised version, we have changed clarification in page 8 line 46 to explain this.
- (21)Comment:** Line 238-239: I do not understand, please explain better.
Response: In the revised version, we have modified the sentence description to make the expression more clear. (page 8 lines 47-48)
- (22)Comment:** Line 254-255: This has not to do with the range of motion but with the segment

dimensions

Response: Thanks for your correction. It has been revised in the updated version at this point (page 9 line 7).

(23)**Comment:** Figure 4: 'Obsavation' -- > observation

Response: Thank you for pointing out the typo. It has been corrected in the revised version. (fig.7)

Responses to reviewer 3

Reviewer Comments: The present work aimed at proposing a real-time motion capture approach with a new regression network architecture for the position of key joints with human shape input and a special inverse kinematic solver based on the human kinematic tree. Results demonstrate that the method is able to improve the inference speed and reduce the latency while ensuring the reconstruction accuracy.

Major comments: The topic of this work is original, causing a probable interest in the reader. However, some changes are needed to improve the clarity of presentation and the comprehensibility of the study.

(1) **Comment:** The abstract contains acronyms which are presented later in the text, but it is the first part to read. It would be advisable to define them also inside the abstract.

Response: Thanks a lot for all your valuable comments. After revision, we have removed acronyms in the abstract, and in the main text, these abbreviations are explained when they first appear.

(2) **Comment:** The introduction is well-written, even if the aim can be stated more clearly. Moreover, a brief description of IMUs with advantages and disadvantages especially related to optical systems is missing. Authors could consider these references to find useful details:

-Digo, E., Pastorelli, S., & Gastaldi, L. (2022). A Narrative Review on Wearable Inertial Sensors for Human Motion Tracking in Industrial Scenarios. *Robotics*, 11(6), 138.

- Filippeschi, A., Schmitz, N., Miezal, M., Bleser, G., Ruffaldi, E., & Stricker, D. (2017). Survey of motion tracking methods based on inertial sensors: A focus on upper limb human motion. *Sensors*, 17(6), 1257

Response: In the revised version, we have added a discussion on the combination of IMU and optical methods in the introduction, and cited the references. (page 1 lines 18-21)

(3) **Comment:** In addition, authors describe their model as divided in two stages, but in Figure 1 there are three blocks with three different colors (body shape estimation, pose estimation and motion reconstruction). I suggest clarifying this aspect in the text, better describing the process proposed in the figure. I would delete lines 84-85, because they represent a comment about

results typical of the discussion section.

Response: Thank you for addressing the unclear presentation in the paper. The two stages mentioned in the paper correspond to the pose estimation stage in Figure 1. In the revised version, we provide explanations that the two stages are included in the pose estimation block of Fig.1 in the page 3 lines 13-14. And We have revised Figure 6, the pipeline diagram, to align with the three stages described in Figure 1. Additionally, we have deleted lines 84-85 from the original manuscript.

- (4) **Comment:** At the beginning of methods, authors should list where they positioned IMUs for a better clarity. Check the equations, because not all of them are cited in the text. I also suggest changing the title of subsection 3.2. because “Motivation” is not so clear.

Response: We have conducted a thorough check of the formulas and ensured that they are properly referenced in the revised version.

In Section "Methods," we have added the following statement at the beginning: "The layout scheme is shown in Fig. 1," which allows readers to refer to Figure 1 to examine the IMUs placement. (page 8 lines 24). Additionally, we have added textual description in the main text (page 8 lines 25-26).

Additionally, we have changed "Motivation" to "Key ideas" to improve clarity in the presentation. (page 9 line 1)

- (5) **Comment:** Results are deeply and consistently discussed.

Response: Thanks.

- (6) **Comment:** Minor comments:

- Line 3: I would define the motion capture as a technique instead of a technology
- Line 5: explain the acronyms VR and AR
- Line 71: explain the acronym SMPL
- Line 76: explain the acronym DNN
- Line 282: explain the acronym MLP

Response: We have changed the description (page 1 line 8). Additionally, we have provided explanations for all acronyms. (page 1 line 9, page 2 line 19, page 3 lines 24-25, page 5 lines 17-18)

REVIEWER COMMENTS

Reviewer #1 (Remarks to the Author):

I appreciate the corrections and explanations provided. The tests performed also look impressive.

The structure of the descriptions and the language raise my doubts. There are some formulations, that indicate that the text does not have the appropriate scientific sound adapted to the requirements of a high-rated journal. The language should be clear, formal, and technical.

Specific notes:

- 1) The abbreviation FIP appears in the abstract and is not explained.
- 2) Sentence 2-3 (page 2) is not entirely true. The proposed method needs calibration (as mentioned in A.5.1).
- 3) Line 40 (page 3) - explain the beta parameter.
- 4) Line 49 (page 3) - what is ground truth? How the real position is obtained? This should also be explained when presenting the results.
- 5) Paragraph from rope 43 (page 3) - Why are errors not counted in one coordinate system?
- 6) The description of the methods used in the comparison is very poor. Care should be taken to clearly link the description with the name of the method used later in the work.
- 7) Line 8 (page 4) - explain offline and online?
- 8) Line 19 (page 4) - what does acceleration optimization mean?
- 9) Table 3 requires a more detailed description. What does "average shape" mean?
- 10) Line 15 (page 7) - what does locally mean?
- 11) Discussion section - it must be honestly admitted that this method will also not be sufficient for skeleton animations. Artifacts are visible and certainly do not meet the current requirements for animation quality (even in mobile games or virtual reality). Unfortunately, in VR, artifacts will also reduce immersion.
- 12) Line 18 page 8 - missing formal definition of systems (origin and axes)?
- 13) Line 31 page 8 - explain "normalized data of IMU"? How? Is it about normalizing the vector length or normalizing all samples? How is rotation normalized?
- 14) Line 46 page 8 - "acceleration and rotation in the body coordinate system" - if so, the input is not raw readings from the sensor, but converted to the body coordinate system. This also needs to be clearly stated. Not every sensor returns an estimated orientation, sometimes it is just measurements in the sensor coordinate system!
- 15) There are 3 different orientation representations in paper - matrix, quaternion, and 6D representation. It should be clearly defined when which representation is used. Why isn't this consistent?

Reviewer #2 (Remarks to the Author):

Dear authors. Thanks for the detailed replies and adaptations made to the manuscript. I do think the presented work has value and I hope to accept the paper after a next review round. I still have some concerns that I would like to be addressed:

- 1) Related to my previous comment 2 on the Inverse Kinematic solver. I understand better now the reason for the original text. **However, I feel like you should emphasize the contribution.** The reason for a neural net to map from joint positions to rotation matrix is because the mapping is redundant. You chose to solve this redundancy by having a neural network and then supervise with data. And yes indeed, if you want to do gradient descent the neural network needs to be differentiable. One could also choose a specific IK solution (in a differentiable way) by f.e. having a predefined rotation axis. That would not be a good solution for sure. **Your contribution is the specific choice of splitting up the IK vs having one network that immediately maps from joint locations to joint rotations (or from IMU readings to joint rotations).** In short, say that IK is redundant and that is why you use a neural network that is supervised by data.
- 2) The ablation on the MPL IK solver is not completely clear to me. What does the following mean: "To meet the conditions of the kinematic inverse solution module based on just a single frame, ..."
- 3) The ablation of shape. Does this mean you inference the network without shape input (i.e. set to zeros), or do you retrain the network without shape as input?
- 4) Related to comment 5. I understand that the device makes a difference, thanks for clarifying. Related to future frames in TIP, please refer to section A of the appendix of the TIP paper: "In practice, filtering causes latency during real-time inference, as computing moving average requires future IMU readings. We use 5 times steps (83ms) of future readings, the same requirement as [Huang et al. 2018; Yi et al. 2021], though they require future readings as part of model input while we merely use them for filtering." Maybe there is some misunderstanding because the TIP paper says they use them 'merely for filtering' which is a bit misleading because one could interpret their moving average filter as a part of the model. **I want to ask the authors to explain this correctly in the text and also adapt the results (this makes FIP look even better).**
- 5) In the section "model training", what is meant with finetuning on the DIP-IMU dataset. This should be reported in more detail to ensure a fair comparison to PIP, TIP etc. My worry would be that FIP becomes very good at the motions included in the DIP dataset, as all 10 participants roughly did the same motion. **I recommend also validate on TotalCapture's version using the real IMU signals (as PIP and TIP do). If the PIP implementation is followed for this validation this could be done without too much effort.**
- 6) The text can still be improved a lot. This is important to make the paper easier to understand. I would encourage the authors to take the time to improve the wording, clarity and correctness of each phrase.

Dear Editor and Reviewers,

Thank you very much for your valuable comments and suggestions to our manuscript. We have carefully revised and improved our manuscript according to your kind advices and detailed suggestions. And the main revisions have been highlighted in manuscript and the revised supplement. The point-by-point response please see the below.

.....

Responses to reviewer 1

Reviewer Comments:

I appreciate the corrections and explanations provided. The tests performed also look impressive.

The structure of the descriptions and the language raise my doubts. There are some formulations, that indicate that the text does not have the appropriate scientific sound adapted to the requirements of a high-rated journal. The language should be clear, formal, and technical.

Response: Thanks a lot for your comments. We have contacted the AJE company to provide language polishing for our manuscript (as shown below). We hope that this will make the paper easier to understand.

[REDACTED]

Specific notes:

- 1) **Comment:** The abbreviation FIP appears in the abstract and is not explained.
Response: The abbreviations have been placed after the first occurrence of their full names.

- 2) **Comment:** Sentence 2-3 (page 2) is not entirely true. The proposed method needs calibration (as mentioned in A.5.1).
Response: The point emphasized here is that there is no longer the need to spend so much time calibrating the relative positions of a large number of sensors and skeletons, which typically required actions such as Apose, Spose, and moving specific distances, often taking around 10 minutes. As shown in A.5.1, the transformation now only needs to be initiated in the T-pose, which takes less than 10 seconds.

The previous statement may be misleading, we have changed the description (lines 1-2, page2).

- 3) **Comment:** Line 40 (page 3) - explain the beta parameter.

Response: Beta is a parameter in SMPL that characterizes the body shape, obtained as a 10-dimensional shape parameter through PCA. The specific computation process for the beta parameters is described in the second paragraph of the Model Formulation section in the paper 'Matthew Loper et al., SMPL: A Skinned Multi-Person Linear Model.'

To prevent any reader misunderstanding, the original text's 'human body' has been specified as the 'SMPL human body model". (line 44, page3)

- 4) **Comment:** Line 49 (page 3) - what is ground truth? How the real position is obtained? This should also be explained when presenting the results.

Response: Ground truth refers to the real values, and here it refers to the real rotations of SMPL joints. We have added expression in lines 2 (page4).

Since the algorithm comparison is conducted using tests on publicly available datasets, these datasets provide corresponding real rotations. By using the official SMPL mapping parameters, the real positions of joints and mesh nodes can be calculated. This information is supplemented in lines 42-43 (page 3).

- 5) **Comment:** Paragraph from rope 43 (page 3) - Why are errors not counted in one coordinate system?

Response: The original description in the article pertains to the calculation method used during experiments. In fact, due to the consistent scale of the coordinate system, these errors have the same values in different coordinate systems. To avoid confusion for readers, these errors are uniformly described as in the body coordinate system. In line 48 (page3), we have revised global as body.

- 6) **Comment:** The description of the methods used in the comparison is very poor. Care should be taken to clearly link the description with the name of the method used later in the work.

Response: To make it clearer, we emphasized the description of the methods used in the comparison, such as DIP, PIP, TIP, and added their respective authors in the introduction section. (line 22, page2 to line 1, page 3)

Moreover, we made revisions to some corresponding descriptions in the ablation study. (lines 2-13 page 7)

- 7) **Comment:** Line 8 (page 4) - explain offline and online?

Response: 'Offline' refers to the non-real-time post-processing of motion reconstruction based on data collected over a certain period of time, which may involve data beyond the current time. In contrast, 'online' denotes real-time motion reconstruction utilizing data available up to the current moment. Additional details have been provided in lines 13-14, page 4.

- 8) **Comment:** Line 19 (page 4) - what does acceleration optimization mean?
Response: Acceleration optimization is presented from "Yi. Physical Inertial Poser (PIP): Physics-aware Real-time Human Motion Tracking from Sparse Inertial Sensors," (Section 3.2.4)
Perhaps our original expression can be misleading, and "acceleration optimization" has been modified to "dynamic optimization of accelerations" to better align with the description in the PIP paper. (line 25, page 4)
- 9) **Comment:** Table 3 requires a more detailed description. What does "average shape" mean?
Response: "Average" refers to setting the shape parameters of SMPL to zero to get the body parameters of the input (height, arm length, and leg length), and additional clarification has been provided in lines 12-13, page 7 of the revision.
- 10) **Comment:** Line 15 (page 7) - what does locally mean?
Response: "Locally" refers to running the method on a typical PC (such as a computer equipped with an Intel i7 processor CPU and an RTX 2060 GPU), which can also be considered as "running on the local machine." The text has been modified in the original document for better understanding. (lines 2-3 page 8)
- 11) **Comment:** Discussion section - it must be honestly admitted that this method will also not be sufficient for skeleton animations. Artifacts are visible and certainly do not meet the current requirements for animation quality (even in mobile games or virtual reality). Unfortunately, in VR, artifacts will also reduce immersion.
Response: We have added restriction on skeleton animations in line 13, page 8. Yes, just as you pointed out, our method has still limitations in many application scenarios and needed to be improved in the further research.
In this paper, our primary emphasis is on highlighting the advancements of our method over existing approaches in certain aspects. Especially in the domain of human-machine interaction analysis, such as the assessment of assembly processes, this level of accuracy is sufficient for analyzing action sequences (e.g., the clockwise or counterclockwise rotation of a screwdriver, or the specific direction in which a hammer is used), as these actions directly impact the correctness of the assembly process. And we also hope this study can provide a new insight for researchers in the field of motion capture using sparse sensors in their subsequent studies.
- 12) **Comment:** Line 18 page 8 - missing formal definition of systems (origin and axes)?
Response: In fact, obtaining more joint movements or rotations in both the body coordinate system and the global coordinate system is beneficial. However, generally, sensors provide information primarily related to the global coordinate system, which implies that it will be easier to achieve results in the global coordinate system.
This has been supplemented in lines 23-24, page 8. Additionally, we have provided textual descriptions for the coordinate system definitions in lines 33-34, page 8.

13) **Comment:** Line 31 page 8 - explain "normalized data of IMU"? How? Is it about normalizing the vector length or normalizing all samples? How is rotation normalized?

Response: 'IMU data normalization' is described in the original text's "Implementation Details." It refers to the process of transforming sensor signals into the body coordinate system. To enhance readability, a reference has been included in lines 38-39, page 8.

14) **Comment:** Line 46 page 8 - "acceleration and rotation in the body coordinate system" - if so, the input is not raw readings from the sensor, but converted to the body coordinate system. This also needs to be clearly stated. Not every sensor returns an estimated orientation, sometimes it is just measurements in the sensor coordinate system!

Response: The description here is indeed not precise. It should be "normalized IMU data." The modification has been made at line 2, page 9.

15) **Comment:** There are 3 different orientation representations in paper - matrix, quaternion, and 6D representation. It should be clearly defined when which representation is used. Why isn't this consistent?

Response: Quaternion form is only used when reading sensor data because the raw output format of the sensor data of our experiment is in quaternion form. Matrix is used as inputs to the model because quaternion is not a continuous representation of rotation. Using quaternion as input would lead to significant jitter in the reconstructed results and could introduce ambiguities in some motions. The 6D representation is used as the output of the model because research has shown that using 6D representation yields better results in human pose estimation compared to matrix and quaternion outputs (as mentioned in "Yi. On the Continuity of Rotation Representations in Neural Networks"). In addition, in the derivations within the manuscript, rotation is represented solely by bold uppercase letter "R". The 6D representation appears in the manuscript as " $R^{(6D)}$ ". The term "quaternion" is not explicitly represented by a letter in the manuscript.

To avoid reader confusion, a description of when quaternions are used has been included in the manuscript at lines 8-10, page 15. And we explained the reasons for using the 6D representation (lines 18-19, page 14).

.....

Responses to reviewer 2

Dear authors. Thanks for the detailed replies and adaptations made to the manuscript. I do think the presented work has value and I hope to accept the paper after a next review round. I still have some concerns that I would like to be addressed:

1) **Comment:** Related to my previous comment 2 on the Inverse Kinematic solver. I understand better now the reason for the original text. **However, I feel like you should emphasize the contribution.** The reason for a neural net to map from joint positions

to rotation matrix is because the mapping is redundant. You chose to solve this redundancy by having a neural network and then supervise with data. And yes indeed, if you want to do gradient descent the neural network needs to be differentiable. One could also choose a specific IK solution (in a differentiable way) by f.e. having a predefined rotation axis. That would not be a good solution for sure. **Your contribution is the specific choice of splitting up the IK vs having one network that immediately maps from joint locations to joint rotations (or from IMU readings to joint rotations).** In short, say that IK is redundant and that is why you use a neural network that is supervised by data.

Response: Thanks a lot for your valuable comments. As you mentioned, our work introduces a new IK solver to address the issue of gradient propagation, and ablation study confirm the effectiveness of this IK solver. In fact, just as you pointed out, we attempted several alternative approaches that were less promising, which led to the choice of the current IK solver design. The contribution was mentioned in the third point of the contributions in the original manuscript but it might not have been well understood. To emphasize this further, the shortcomings of existing IK methods and our contributions have been provided at the lines 26- 27, page 3.

- 2) **Comment:** The ablation on the MPL IK solver is not completely clear to me. What does the following mean: “To meet the conditions of the kinematic inverse solution module based on just a single frame, ...”

Response: This is in contrast to other methods, such as DIP (Huang et al. 2018), Transpose (Yi et al. 2021), etc., where the process of mapping joint positions to joint rotations is based on multiple frames (bidirectional RNN). Using the MLP IK solver refers to replacing the current IK design with a regular MLP and retraining to obtain a new model. We have provided further details at lines 4-5, page7.

- 3) **Comment:** The ablation of shape. Does this mean you inference the network without shape input (i.e. set to zeros), or do you retrain the network without shape as input?

Response: Yes, the latter you mentioned is right. This involves retraining the network after removing the shape (skeleton) input branch of pipeline. To make it clearer, we emphasized it at lines 2, page 7.

- 4) **Comment:** Related to comment 5. I understand that the device makes a difference, thanks for clarifying. Related to future frames in TIP, please refer to section A of the appendix of the TIP paper: “In practice, filtering causes latency during real-time inference, as computing moving average requires future IMU readings. We use 5 times steps (83ms) of future readings, the same requirement as [Huang et al. 2018; Yi et al. 2021], though they require future readings as part of model input while we merely use them for filtering.” Maybe there is some misunderstanding because the TIP paper says they use them ‘merely for filtering’ which is a bit misleading because one could interpret their moving average filter as a part of the model. **I want to ask the authors to explain this correctly in the text and also adapt the results (this makes FIP look even**

better).

Response: Yes, perhaps this could indeed lead to some misunderstanding, as we didn't find the input of future frames when reviewing the TIP code. After consulting the TIP appendix, we did indeed discover the 83ms latency you mentioned. We have made corrections in the Tab. 1 and provided textual explanations at lines 8-9, page 4. Thanks a lot for your reminder.

- 5) **Comment:** In the section “model training”, what is meant with finetuning on the DIP-IMU dataset. This should be reported in more detail to ensure a fair comparison to PIP, TIP etc. My worry would be that FIP becomes very good at the motions included in the DIP dataset, as all 10 participants roughly did the same motion. **I recommend also validate on TotalCapture’s version using the real IMU signals (as PIP and TIP do). If the PIP implementation is followed for this validation this could be done without too much effort.**

Response: Thanks a lot for your suggestion. We ensure consistency in the division of the training and test sets for DIP-IMU with other methods, and we have made adjustments to the relevant statements (line 19, page 15).

We conducted experiments once again on the TotalCapture dataset, and the comparisons of certain metrics are as follows:

Method	Aang	Joint	Mesh	Jitter
DIP	17.22	9.42	11.22	3.62
Transpose	12.89	6.55	7.42	0.95
PIP	12.04	5.61	6.51	0.20
TIP	13.55	5.65	6.78	0.87
FIP(ours)	14.47	5.92	8.04	1.83

It can be observed that our method performs relatively well on Totalcapture dataset without concerns of overfitting. However, for the following reason, we believe it is better not to include this result in the main text of the paper but in the Section B of the Supplement:

The TotalCapture dataset does not provide any body shape information (while DIP-IMU dataset provide the height/weight/sex to estimate the shape), so we can only set the SMPL shape parameters to 0, which reduces the method's accuracy, as mentioned in lines 39-40, page 3. Therefore, including it in the paper may not be appropriate, which affects the consistency of the results.

- 6) **Comment:** The text can still be improved a lot. This is important to make the paper easier to understand. I would encourage the authors to take the time to improve the wording, clarity and correctness of each phrase.

Response: Thank you for your suggestions. We have made revisions to certain expressions and sentences that might have been misleading based on some of the feedback. And we have also contacted the AJE company to provide language polishing for our manuscript (as shown below). We hope that this will make the paper easier to understand.

[REDACTED]

REVIEWER COMMENTS

Reviewer #1 (Remarks to the Author):

Unfortunately, the corrections do not fully meet my expectations and introduce further doubts.

Page 2, lines 1-2 - Please provide the source of this information. What systems take more than 10 minutes to calibrate?

Explain and prove the goal of the work "2. enhancing the expressive power of neural network structures;"

Table 1 and 2 – These are mean values, you should also give standard deviation. Is this the average value for all recordings (and all frames in one recording) in the database?

I also have doubts about the statement ground true. What do you mean by real rotations (page 4, line 2)? Is that rotation measured (by what?) or estimated by SMPL? If estimated or measured give also information on that method error.

IMU data Normalization is very confusing (page 15 and page 8). Still, there is no explanation (with an equation) of calibration (especially on page 15).

Equations 11 and 12 can also be written in quaternion form. I don't fully agree with the justification for using three 3D rotation representations. If the input of the model is the R (matrix) and the output is R^6D . So where is the transformation? As described in the "Tree-like Inverse Kinematic Solver" section, the rotation matrix R is also used here.

There are no details related to the training (page 15), what parameters were achieved? On what set (validation, training)?

There seems to be a lot of DIP elements at work. Please explain the differences and new features more.

In the paper about DIP (Table 7, <https://arxiv.org/pdf/1810.04703.pdf>) we can find very good performance results. Please comment according to your sentence about method performance (page 4).

Reviewer #2 (Remarks to the Author):

Thanks for your clear answers and clarifications in the text.

In short, I think the paper is ready to be published now.

Great that you did additional tests on TotalCapture that show very reasonable results. (The jitter metric of PIP seems artificially low, and I have noticed that before myself as well.)

If you'd still want to include your TotalCapture results in the paper, you can get the shape parameters from each subject by looking into to the TotalCapture split of AMASS where the shape is provided. I think it is fine if you do not do this, but maybe you will come closer to PIP results.

Congratulations on this nice work!

Dear Editor and Reviewers,

Thank you very much for your valuable comments and suggestions to our manuscript. We have carefully revised and improved our manuscript according to your kind advices and detailed suggestions. And the main revisions have been highlighted in manuscript and the revised supplement. The point-by-point response please see the below.

.....

Responses to reviewer 1

Unfortunately, the corrections do not fully meet my expectations and introduce further doubts.

Comment: Page 2, lines 1-2 - Please provide the source of this information. What systems take more than 10 minutes to calibrate?

Response: 10 minutes is an empirical value we obtained through experimentation, based on our statistics using the Noitom PN3. The time is mainly spent on maintaining and executing various poses, including not only the T-pose but also A-pose, S-pose, B-pose, etc., all of which require time. In addition, the system also requires users to move a certain distance for calibration.

Additionally, the observation that setting up dense sensors is more time-consuming compared to sparse sensors is also mentioned in the introduction section of the DIP paper: "placing 17 sensors ... can be intrusive, time-consuming...". Perhaps it includes the time spent on wearing, however, it was found that calibration also takes more time. This is because calibration with sparse sensors can be completed with just T-pose, which is also mentioned in Appendix A of Tanspose (<https://arxiv.org/pdf/2105.04605.pdf>.)

Anyway, the specific time comparison is indeed not rigorous enough. In the revision, we modified the specific time comparison to a comparison of the complexity of calibration, as "The calibration process of sparse sensors is simpler than that of dense sensors." (line 1, page 2)

Comment: Explain and prove the goal of the work "2. enhancing the expressive power of neural network structures;"

Response:In our paper, the enhancement of expressive power is validated through the reduction in errors, as was done in the paper "The Expressive Power of Neural Networks: A View from the Width, <https://arxiv.org/pdf/1709.02540.pdf>", and the improvement in the expressive power of FIP structure mainly stems from the following aspects:

1. Considered the influence of human body shape parameters on the fitting results;
2. Proposed a new kinematic inverse solver;
3. Utilized a shared model for different sensors.

These improvements were compared in the ablation study presented in Table 3 of the paper. The results indicate that these designs effectively reduce errors. These designs are incorporated into the neural network structure of FIP, indicating that these designs contribute to enhancing the expressive power of the neural network structure.

To make it clearer, we modified it as: 2. enhancing the expressive power of neural network

structures by considering the human body shape parameters, using a new kinematic inverse solver and a shared model for different sensors. (lines 18-19, page 3)

Comment: Table 1 and 2 - These are mean values, you should also give standard deviation. Is this the average value for all recordings (and all frames in one recording) in the database?

Response: Table 1 principally exhibits metrics related to accuracy. In the latest two works, TIP (as shown in Table 1 of <https://arxiv.org/pdf/2203.15720.pdf>) and PIP (as shown in Tables 1, 2, and 3 of <https://arxiv.org/pdf/2203.15720.pdf>), only the mean values are presented without accompanying standard deviations. To align with these works, our paper likewise displays the mean values.

However, we recognize that without standard deviations, it is difficult to depict the distribution of errors and to substantiate the efficacy of the methods. Therefore, drawing inspiration from PIP's approach, we provided error distribution graphs (Fig.2) corresponding to the four precision metrics in Table 1. These graphs offer a more intuitive reflection of the distribution of errors than standard deviations.

Table 2 exhibits the deployment metrics for embedded devices. It is not feasible to calculate the standard deviation for GPU memory usage (MEM), as MEM does not change during runtime. This is attributable to PyTorch's method of allocating GPU memory, which remains stable throughout the duration of execution. Similarly, it is difficult to compute the standard deviation for other deployment-related metrics (TPF, FPS, latency) since these values generally do not fluctuate during stable operation. The variance in these metrics depends on the operating state of the device and is unrelated to the method itself, such as whether other code is being executed concurrently or large programs are running. The experiments presented in our paper ensured consistency in the operating state.

Yes, this is the average value across all frames for all recordings in the test set, which is explained in section model training (lines 21-24, page 15) and is consistent with the compared methods (section 4.3 of Transpose, section 5.1.2 of DIP paper, section 4 of PIP, etc.).

Comment: I also have doubts about the statement ground true. What do you mean by real rotations (page 4, line 2)? Is that rotation measured (by what?) or estimated by SMPL? If estimated or measured give also information on that method error.

Response: "Ground truth" is used to represent the true values. The term is mentioned at: section 5.2 of DIP (<https://arxiv.org/pdf/1810.04703.pdf>), section 4.2.2 of Transpose (<https://arxiv.org/pdf/2105.04605.pdf>) and section 4.1 of TIP (<https://arxiv.org/pdf/2203.15720.pdf>).

This paper uses the same term to denote true values, maintaining consistency with these works.

The "real rotations" are provided directly by the dataset. For instance, the AMASS dataset offers 72-dimensional SMPL parameters, which includes axis-angle rotations for each joint. Similarly, the DIP-IMU dataset provides joints' rotations directly. It does not

require any additional devices or methods to estimate the real rotation. At this point, FIP maintains consistency with other papers such as DIP, Transpose, PIP, and TIP. And we have revised it as: the real rotations of the SMPL joints directly provided by the dataset. (lines 2-3, Page 4)

Comment: IMU data Normalization is very confusing (page 15 and page 8). Still, there is no explanation (with an equation) of calibration (especially on page 15).

Response: Normalization process is consistent in past studies. Section 4.3 of the DIP and section A.2 of the appendix of the Transpose contain almost the same form as the mentioned equations. The process of data normalization is described in equation 11) 12) of the article. This process is to transform the sensor signal from the world coordinate system to the human body coordinate system. We explained further in the model training section. (lines 9-10, page 15)

In the section Result, we uses a publicly available dataset, and the DIP-IMU and TotalCapture dataset have already provided calibrated sensor data.

in section A.5.1 of the supplement (lines 107-108), we explained the calibration process of the demo and provided the equations.

Comment: Equations 11 and 12 can also be written in quaternion form. I don't fully agree with the justification for using three 3D rotation representations. If the input of the model is the R (matrix) and the output is R^{6D}. So where is the transformation? As described in the "Tree-like Inverse Kinematic Solver" section, the rotation matrix R is also used here.

Response: As you pointed out, Equations 11 and 12 can also be written using quaternions, but only rotation matrices R and R^{6D} are used in the model computation. The quaternion mentioned has no direct involvement in the model computation.

In the section "Tree-like Inverse Kinematic Solver," rotation matrices R have been used to demonstrate the design principles of the IK solver in this paper. It should be noted that the R in the formulas is not equivalent to the model output, which remains R^{6D}.

The transformation between R^{6D} and R is as follows, and the clarification on the transformation has been added to the section C.1 of the supplement of the revision.

From rotation matrix to R^{6D} of the ground truth:

$$\mathbf{R}_{\text{GT}} = \begin{bmatrix} r_{11} & r_{12} & r_{13} \\ r_{21} & r_{22} & r_{23} \\ r_{31} & r_{32} & r_{33} \end{bmatrix} \quad (5)$$

$$\mathbf{R}_{\text{GT}}^{(6D)} = [r_{11} \ r_{21} \ r_{31} \ r_{12} \ r_{22} \ r_{32}]^T \quad (6)$$

From R^{6D} to rotation matrix of the prediction:

$$\mathbf{R}_{\text{pred}}^{(6D)} = [r'_1 \ r'_2 \ r'_3 \ r'_4 \ r'_5 \ r'_6]^T \quad (7)$$

$$r_{\text{norm}}[1] = \|[r'_1 \ r'_2 \ r'_3]^T\|_2 \quad (8)$$

$$r_{\text{norm}}[2] = \|[r'_4 \ r'_5 \ r'_6]^T\|_2 \quad (9)$$

$$\begin{bmatrix} r'_7 \\ r'_8 \\ r'_9 \end{bmatrix} = \frac{[r'_1 \ r'_2 \ r'_3]^T}{r_{\text{norm}}[1]} \times \frac{[r'_4 \ r'_5 \ r'_6]^T}{r_{\text{norm}}[2]} \quad (10)$$

$$\mathbf{R}_{\text{pred}} = \begin{bmatrix} r'_1/r_{\text{norm}}[1] & r'_4/r_{\text{norm}}[2] & r'_7 \\ r'_2/r_{\text{norm}}[1] & r'_5/r_{\text{norm}}[2] & r'_8 \\ r'_3/r_{\text{norm}}[1] & r'_6/r_{\text{norm}}[2] & r'_9 \end{bmatrix} \quad (11)$$

Comment: There are no details related to the training (page 15), what parameters were achieved? On what set (validation, training)?

Response: Page 15 details various conditions and parameters of the training process, such as learning rate, batch size, segment length for processing sequence data, the optimizer used, and others. Based on the provided above parameters, it should be possible to faithfully reproduce the training process, and the model structure is also available in the provided code.

The trained and achieved parameters is the weights of the neural network. The well-trained weights can be obtained according to the instructions provided in the code. We have made modifications in the revision to prevent readers from feeling confused. (line 3, page 16)

In the original text, the "model training" section (lines 21-24, page 15) introduced the training approach and the dataset partition. The model is pre-trained on the AMASS dataset and fine-tuned on the DIP-IMU training split.

Comment: There seems to be a lot of DIP elements at work. Please explain the differences and new features more.

Response: DIP introduced deep neural networks to this field, proposed various data processing methods, and provided an open-source dataset, making it an outstanding contribution (lines 20-21, page 2). Subsequent related methods (Transpose, TIP, PIP) have also maintained consistency with it. For instance, they utilised the same datasets, training methods and data preprocessing techniques to ensure comparison results were consistent. This paper adopts the same approach.

Compared with DIP, FIP differs significantly in terms of model design.

1. It can be observed that DIP utilizes only one RNN to establish the mapping relationship from sensor signals to rotations. In contrast, this paper employs three RNNs, each dedicated to fitting the displacement of leaf nodes, the position of upper-body SMPL nodes, and the position of lower-body SMPL nodes, respectively.
2. In addition to the same element of using RNN, FIP incorporates the following design aspects: (1) FIP takes into consideration the utilization of human body shape to predict

the human body SMPL skeleton, which is integrated as a novel feature input into the Leaf regressor, upper/lower body regressor, and Inverse Kinematic Solver. (2) FIP also introduces a new Inverse Kinematic Solver. These design choices have been experimentally validated through ablation studies, confirming their effectiveness.

Comment: In the paper about DIP (Table 7, <https://arxiv.org/pdf/1810.04703.pdf>) we can find very good performance results. Please comment according to your sentence about method performance (page 4).

Response: As you mentioned, DIP did indeed yield the best performance results at that time. As shown in Table 7 of the DIP paper, the fastest running speed is achieved when using the RNN, and the introduction of **BiRNN** significantly reduces the running speed (the fastest speed is around 30 FPS). Moreover, in Table 2 of the DIP paper, it can be seen that **BiRNN** achieves the highest reconstruction accuracy.

Therefore, to validate the accuracy of FIP, we compares it using the **BiRNN** model.

Due to the utilization of 5 future frames in DIP, the **latency** of DIP is at least 85 ms with the sensor outputs data at 60 FPS (section7 in the DIP paper). Moreover, the experiments of DIP were conducted on a remote laptop. Generally, laptops exhibit greater computational power than embedded devices.

Anyway, DIP has made a great contribution to the research in this field. Over the past five years, many other methods have built upon DIP as a benchmark. For example, Transpose introduced translation estimation, PIP enhanced accuracy estimation, TIP added terrain estimation, and our method, FIP, increased computational speed. Additionally, it is noteworthy that DIP utilized an older version of the TensorFlow library, which differs from the PyTorch library employed by subsequent methods. This difference may also contribute to a difference in computational performance.

We have revised the sentence to provide a more impartial evaluation of the deployment performance of DIP. (line 17, page 4)

REVIEWERS' COMMENTS

Reviewer #1 (Remarks to the Author):

I don't entirely agree with the explanations. I think it's enough to explain and make the corrections in the text of the paper.

Symbols should be unified in the whole work. For example, R^{6D} , $R^{(j+1)}_j$, R^{-1}_{root} . The meaning of index varies.

As I understood the explanations:

"Alignment with the global coordinate system" - this is described as calibration.

To convert the sensor signal from the world coordinate system to the human body coordinate system - this is described as normalization.

In my opinion, it is the same... and it should be rather the calibration. There may also be inaccuracies in referenced publications. Normalization has its definition in the sense of statistics.

I have the same reservation about "ground truth". In the publication, this is the basis for calculating the errors. So You need to explain what this means. This value must have been determined or measured somehow (this is probably explained in the publications regarding the used databases). This needs to be explained. What does this value mean in the given application? Regarding training (page 15), quantitative data specifications (training and testing) are still missing. What training parameters were obtained (loss, accuracy..., etc.) that determined the end of the process.

Reviewer #4 (Remarks to the Author):

Paper summary:

This work tackles the problem of motion capture from sparse IMUs mounted on the human body. More precisely, this work tackles the problem of efficiently recovering the body pose given limited compute hardware. To this end, they adopt neural network architectures such that their design is reflecting the structure of the human skeleton. Their results show that this design is superior to other alternatives. Moreover, they propose a kinematics solver, which recovers the joint angles from the positional estimates of the neural networks.

Strengths:

The problem setting and method is sound.

The manuscript contains many visualizations, which support their claims and provide additional insights for the reader.

I generally like the idea of incorporating information of the body shape into the motion capture process. In particular, the idea of thinking about "minimal" and easy-to-perform measurements, as the authors propose, are interesting and very relevant for real-world applications.

The results demonstrate an improvement over the state of the art, especially in terms of runtime and latency, which was the main goal of this work.

The authors share their code, which can help to further stimulate research in this direction.

Weaknesses:

I like the Key Ideas section as it conceptually explains the insights. However, the Pipeline and Tree-like Inverse Kinematics Solver lacks such a "high-level intuition". Why is this pipeline different from

prior designs? Why is this design superior? It would be nice to add such discussions about design alternatives into this paragraph. To be clear, I can see the differences in the figure, however, they are not explained in text. The same applies to the kinematics solver.

The Pipeline paragraph would also benefit from more explanations. While the inverse kinematics part is quite extensively explained, the positional regression step lacks more in-depth discussion.

Final comments:

Overall, I believe the strengths of this work outweigh the weaknesses. Moreover, I believe the weaknesses can be addressed for the camera-ready version and I would encourage the authors to do so. Regarding the previous concerns and the authors' rebuttal, I believe the answers are convincing and sufficiently addressed in the new manuscript version.

Minor:

Missing citations (no comparison needed since this work assumes a body-mounted camera in addition to IMUs):

```
@article{EgoLocate2023,  
author = {Yi, Xinyu and Zhou, Yuxiao and Habermann, Marc and Golyanik, Vladislav and Pan,  
Shaohua and Theobalt, Christian and Xu, Feng},  
title = {EgoLocate: Real-time Motion Capture, Localization, and Mapping with Sparse Body-  
mounted Sensors},  
journal={ACM Transactions on Graphics (TOG)},  
year = {2023},  
volume = {42},  
number = {4},  
numpages = {17},  
articleno = {76},  
publisher = {ACM}  
}
```

Confusing statement on page 11 l.16, why is inverse kinematics not differentiable and can thus not be used for network training? There has been prior work showing this:

```
@inproceedings{deepcap,  
title = {DeepCap: Monocular Human Performance Capture Using Weak Supervision},  
author = {Habermann, Marc and Xu, Weipeng and Zollhoefer, Michael and Pons-Moll, Gerard and  
Theobalt, Christian},  
booktitle = {{IEEE} Conference on Computer Vision and Pattern Recognition (CVPR)},  
month = {jun},  
organization = {{IEEE}},  
year = {2020},  
}
```

Maybe the authors wanted to say something different here, and this statement needs some more refinement.

Typo on page 12 l.10 p_{leaf}

Page 14 l.22 "For the loss of rotation" → this means you loose rotation. I guess you want to say "For supervising the rotations"

Dear Editor,

Thank you very much for your valuable comments and suggestions to our manuscript. We have carefully revised and improved our manuscript according to your kind advices and detailed suggestions. And the main revisions have been highlighted in the revised version manuscript. The point-by-point response please see the below.

.....

Responses to reviewer 1

Comment: Symbols should be unified in the whole work. For example, R^{6D} , $R^{(j+1)}_j$, R^{-1}_{root} . The meaning of index varies.

Response: We have thoroughly reviewed the entire document and made revisions to the use of symbols. Specifically, we enclose expressions with 6D representation in parentheses as superscripts. We have adopted a more standardized notation for matrix inverse, denoted as $[R_{root}]^{-1}$. For symbols with two subscripts, the first subscript indicates the position, and the second subscript indicates the source. For example, $R_{l,s}$ represents the sensor value at the leaf node, and $P_{\{all,gt\}}$ represents the ground truth value for all nodes. Additionally, we have double-checked the entire document to ensure that each symbol is adequately explained in the main text.

Comment: I have the same reservation about "ground truth". In the publication, this is the basis for calculating the errors. So You need to explain what this means. This value must have been determined or measured somehow (this is probably explained in the publications regarding the used databases). This needs to be explained. What does this value mean in the given application?

Response: In real-world applications, "ground truth" represents the true joint rotations or joint positions of a person. In DIP, an iterative optimization approach (SIP) is employed to obtain accurate joint positions on 17 sensors (though errors may still be present). In AMASS, optical marker-based methods are used to acquire precise human body poses. We have added this in the lines 40-41, page3.

Comment: Regarding training (page 15), quantitative data specifications (training and testing) are still missing. What training parameters were obtained (loss, accuracy..., etc.) that determined the end of the process.

Response: The description in the text mentions that training was stopped at the 3k-th epoch. Generally, when the training loss is decreasing while the test loss is increasing, it indicates overfitting of the model to the training set. Therefore, we chose the inflection point of the test loss as the criterion for stopping training. Upon observing the curves during the training process, this point happened to be around the 3k-th epoch. We have supplemented this in the line 15 page 16.

.....

Responses to reviewer 4

Comment: However, the Pipeline and Tree-like Inverse Kinematics Solver lacks such a

“high-level intuition” . Why is this pipeline different from prior designs? Why is this design superior? It would be nice to add such discussions about design alternatives into this paragraph. To be clear, I can see the differences in the figure, however, they are not explained in text. The same applies to the kinematics solver.

Response: We added “high-level intuition” in Section Pipeline (lines 10-13, page 10) in the revision, and a detailed discussion on the superior performance of this method is provided in Section C of the appendix.

Comment: Missing citations (no comparison needed since this work assumes a body-mounted camera in addition to IMUs):

Response: We have added it in the section Introduction (line 25, page2).

Comment: Confusing statement on page 11 l.16, why is inverse kinematics not differentiable and can thus not be used for network training?

Response: The process of inverse kinematics in Euclidean space is non-differentiable. This means that if we use traditional inverse kinematics methods to supervise joint rotations, the gradients cannot propagate backward to the trained network parameters, making the model untrainable. We have revised the statement in lines 20-22, page 11.

Comment: Typo on page 12 l.10 p_{leaf}, Page 14 l.22 “For the loss of rotation” → this means you loose rotation. I guess you want to say “For supervising the rotations”

Response: We have corrected these typos and inappropriate expressions. (line 3, page 13 and line 16, page 14)